# Physically-based modelling of glacier evolution under climate change in the tropical Andes

Jonathan D. Mackay[1,2], Nicholas E. Barrand[2], David M. Hannah[2], Emily Potter[3,4], Nilton Montoya[5], Wouter Buytaert[6]

[1]British Geological Survey, Environmental Science Centre, Keyworth, Nottingham, UK
[2]School of Geography, Earth and Environmental Sciences, University of Birmingham, Edgbaston, Birmingham, UK
[3]School of Geography, University of Leeds, Leeds, UK
[4]Department of Geography, University of Sheffield, Sheffield, UK
[5]Agriculture Department, Universidad Nacional de San Antonio Abad del Cusco, Cusco, Peru
[6]Civil and Environmental Engineering, Imperial College London, London, UK

*Correspondence to*: Jonathan D. Mackay (joncka@bgs.ac.uk)

**Abstract.** In recent years, opportunities have opened up to develop and validate glacier models in regions that have previously been infeasible due to observation and/or computational constraints, due to the availability of globally-capable glacier evolution modelling codes and spatially-extensive geodetic validation data. The glaciers in the tropical Andes represent some of the least observed and modelled glaciers in the world, making their trajectories under climate change uncertain. Studies to date, have typically adopted empirical models of the surface energy balance and ice flow to simulate glacier evolution under climate change, but these may miss important non-linearities in future glacier mass changes. We combine two globally-capable modelling codes that provide a more physical representation of these processes: i) JULES which solves the full energy balance of snow and ice; and ii) OGGM which solves a flowline representation of the shallow ice equation to simulate ice flow. JULES-OGGM is applied to over 500 tropical glaciers in the Vilcanota-Urubamba basin in Peru, home to more than 800,000 people consisting predominantly of rural communities with low socioeconomic development and high vulnerability to climate change. The model is evaluated against available glaciological and geodetic mass balance observations to assess the potential for using the modelling workflow to simulate tropical glacier evolution over decadal timescales. We show that the JULES-OGGM model can be parameterised to capture decadal (2000-2018) mass changes of individual glaciers, but that limitations of the JULES prognostic snow model prevent accurate replication of observed surface albedo fluctuations and mass changes across all glaciers simultaneously. Specifically, the model cannot replicate the feedbacks between the driving meteorology, surface energy balance, ablation processes and snow darkening. Only by forcing the model with observed net radiation variables, were we able to capture observed surface albedo dynamics. When driven with statistically-downscaled climate change projections, the JULES-OGGM simulations indicate that, contrary to point-scale energy balance studies, sublimation plays a very minor role in glacier evolution at the basin scale and does not bring about significant non-linearities in the glacier response to climate

warming. The ensemble mean simulation estimates that total glacier mass will decrease to 17% and 6% of that in 2000 by 2100 for RCP4.5 and RCP8.5 respectively which is more conservative than estimates from some other global glacier models.

**1 Introduction**

Meltwater from tropical Andean glaciers buffers water supply to domestic users, agriculture and for energy supply during periods of drought (Buytaert et al., 2017; Carey et al., 2014; Ultee et al., 2022). The magnitude of buffering will be inhibited in the future as glaciers recede in a warming climate. Large-scale earth observation analyses of historical glacier extent and geodetic mass balance indicate that tropical Andean glaciers have been receding rapidly in recent decades. Dussaillant et al.

(2019) found that between 2000 and 2018, the average mass balance of glaciers in the tropical Andes was -0.42 ± 0.24 m yr−1 w.e. Seehaus et al. (2019) estimated that glacier extent in the Peruvian Andes, home to ~70% of the world's tropical glaciers reduced by 29% between 2000 and 2016 with over half of this retreat occurring in the last three years of this period. Taylor et al. (2022) estimated glacier area losses of between 54 and 64% (1970-2020) across three mountain ranges in the Southern Peruvian Andes.

There is high confidence that temperatures in the Andes will continue to increase throughout the 21st century (Ipcc, 2023; Yarleque et al., 2018). The diversity of meltwater end-users and potential impacts of climate change necessitate reliable projections of glacier mass changes to inform policy and adaptation pathways (Johansen et al., 2018). This is especially true in the tropical Andes where many aspects of future climate change remain highly uncertain, due to inadequate climatic and glaciological monitoring networks.

Providing reliable projections of glacier mass changes for the tropical Andes is challenging due, in part, to the high spatiotemporal variability in ablation processes. At lower elevations below the equilibrium line altitude, ablation is dominated by melt. At higher elevations, point scale energy balance studies show that sublimation can reduce the energy available for melting, particularly in the cooler dry season when humidity is low and the surface roughness of ice is higher (Gurgiser et al., 2013; Winkler et al., 2009). The implications of this for simulating glacier mass change was demonstrated by Fyffe et al.

(2021). They applied an energy balance model at five on-ice meteorological stations in Cordillera Blanca and Cordillera Vilcanota in Peru and perturbed precipitation and temperature inputs to explore the mechanisms of mass balance sensitivity to climate. They showed that, at lower elevations, the mass balance signal was driven by a switch of snow to rainfall. At higher elevations, the change in mass balance was driven by a switch from sublimation to melt processes. The mass balance response to climate warming should, therefore, be expected to be non-linear and spatially variable in response to temperature changes

brought about by climate change. Indeed, Marzeion et al. (2012) concluded that the inability of their temperature index model to capture glacier mass dynamics in the tropics likely stemmed from the fact that it did not account for sublimation processes. Simulating mass changes over periods of decades is further-complicated by the need to account for ice mass redistribution and the corresponding elevation feedback to accumulation and ablation processes (Huss et al., 2010; Van Tiel et al., 2018).

The sparsity of glacier monitoring data and meteorological observation networks in the tropics (Gärtner-Roer et al., 2019) has led many glacier evolution modelling studies to use simplified models that do not necessarily account for potentially important processes (e.g. Somers et al. (2019) did not explicitly model ice flow) or constrain their simulations to discrete on-ice locations where required observation data exist (Fyffe et al., 2021; Rabatel et al., 2013), limiting their usefulness for basin-scale assessments of meltwater perturbations under climate change. However, significant advances in methods to derive glacier mass variations from earth observation data has led to the development of regional and global geodetic datasets of glacier mass change in recent years (Dussaillant et al., 2019; Hugonnet et al., 2021). While they are subject to uncertainty, they provide the means to validate glacier evolution model simulations in regions that have previously been infeasible due to lack of field data. The availability of these data has coincided with the development of a growing number of large-scale (up to global) glacier evolution models (Marzeion et al., 2020). The Open Global Glacier Model (OGGM) is the first open source, "community" model for global glacier modelling (Maussion et al., 2019). It takes advantage of global glacier outline and topographic datasets, automated approaches for glacier geometry delineation and global geodetic and glaciological mass balance data for parameter identification to provide a workflow, through which users can build and run an ice dynamics model of any glacier in the world. The availability of flexible, globally-capable glacier modelling codes like OGGM, and spatially-extensive geodetic mass data allowed Caro et al. (2023) to estimate the contribution of glacier meltwater runoff for 786 glacierised river basins in the Andes: a feat that would not have been possible until recently.

OGGM stands out from many other globally-capable glacier models in that it implements an efficient, physically-based flowline model to simulate ice redistribution rather than an empirical approach, such as volume-area scaling (Bahr et al., 2015), which employ parameterisations that are, arguably, less robust for decadal projections. A potential limitation of OGGM, however, is that it does not include a physical representation of the surface energy balance and, therefore, cannot necessarily capture the non-linearities in ice evolution under climate change that are expected to be observed for glaciers situated in the tropics.

A very different globally-capable model is the Joint UK Land Environment Simulator (JULES), a community land surface model that was originally developed as the land surface component of the Met Office Unified (climate) Model, (Best et al., 2011), but is now used more broadly by environmental scientists as a standalone model to simulate a range of land surface processes. This includes a physically-consistent energy balance and snow pack model that can be used to simulate surface mass balance changes of snow and ice. While it has no representation of ice dynamics, Shannon et al. (2019) demonstrated that the snow and ice component of JULES can be used to simulate the surface mass balance of glaciers. In this study, we present a physically-based glacier modelling workflow that combines the energy balance model in JULES with the ice dynamics model in OGGM. We apply the model to the Vilcanota-Urubamba basin in Southern Peru which contains over 500 tropical glaciers, including the world's largest: Quelccaya and is home to the second-largest tropical glacierised mountain range in the world. The basin is home to over 800,000 people with mostly traditional livelihoods, low socioeconomic development and high poverty. Water is used for irrigated agriculture and hydropower plants which both rely on year-round

runoff from glaciers. The basin also provides drinking water for the densely populated city of Cusco which has experienced severe drought in recent years. We aim to:

1) Assess the model against available geodetic and glaciological mass balance observations and on-ice energy balance observations to explore the potential for and limitations of a physically-based glacier evolution model in a tropical setting.

2) Drive the model with twenty-first century climate simulations to forecast changes in ice mass and area and explore the controls on these changes at the process level.

## 2 Methodology

### 2.1 Study basin

The study focuses on the glacierised Vilcanota-Urubamba basin (VUB), situated in the Cusco region of Southern Peru between the dry high-Andean altiplano and more humid Amazon basin (Figure 1). The VUB is characterised by a complex topography (~1200-6400 m asl.) which is dominated by the three glacierised mountain ranges; the Cordillera Vilcanota in the south-east and the Cordillera Urubamba and Vilcabamba in the north-west. Total glacier area reduced by 37% between 1988-2016 to ~142 km$^2$ (Drenkhan et al., 2018). The remaining ice is situated between 4500-6000 m asl and is predominantly south-facing, but with considerable variation across the basin. Typical glacier slope ranges between 20-40 degrees, but in the Cordillera Vilcanota, a large number of glaciers are perched on the gentler rolling slopes.

The climatology is typical of an outer-tropical setting, with a pronounced wet and dry season (austral summer and winter respectively, Figure 2b), however, hydroclimatology is highly variable within the basin. Mean annual precipitation ranges from ~1000 mm in the drier Cordillera Vilcanota up to ~2000 mm for glaciers on the western edge of the VUB. Temperature is strongly controlled by elevation and the annual temperature range is generally small (~4.5 °C between December and July, Figure 2d).

### 2.2 Glacier outlines, mass balance and volume data

The glacier outlines used in this study were derived by Drenkhan et al. (2018) from multi-spectral optical satellite data for the year 1998. These were chosen over the Randolph Glacier Inventory (RGI version 6 at the time of writing) as they have been ground-truthed and optimised to avoid known issues with seasonal snow cover that impact the RGI (Rgi Consortium, 2023). Glacier intersects were manually determined, resulting in a total of 532 individual glaciers.

Two sources of glacier mass balance data were used in this study. The VUB benefits from glaciological mass balance measurements from an ablation stake network, made up of 13 stakes, on the Suyuparina glacier which is situated within 1-2 km of the Quisoquipina meteorological station. The data span October 2013 to December 2014 at elevations between 5135 and 5190 m asl (Molina et al., 2015) and provide information on within-year mass balance dynamics. In addition, geodetic data were taken from Dussaillant et al. (2019) who generated glacier mass balance estimates over 2000 to 2018 for the entire

Andes mountain range on a 30 m grid using the 'Advanced Spaceborne Thermal Emission and Reflection Radiometer (ASTER) monitoring of ice towards extinction' geodetic method (Brun et al., 2017). These data are provided as mean annual

elevation change rates over the 18-year time period which were converted to mass loss with an assumed ice density of 917 kg m$^{-3}$. They provide information on longer-term mass balance trends and spatial variability which is known to be high in the tropical Andes (Clark et al., 2020).

We also obtained glacier volume data from Millan et al. (2022) who use an ice motion mapping approach to estimate ice thickness and volume of the world's glaciers. These data are subject to some uncertainty given the temporal mismatch between

their ice velocity maps which were recorded for the years 2017 and 2018, glacier outlines based on maps of Drenkhan et al. (2018) and digital elevation model. Nonetheless, they serve as a useful comparison for glacier dynamics modelling studies.

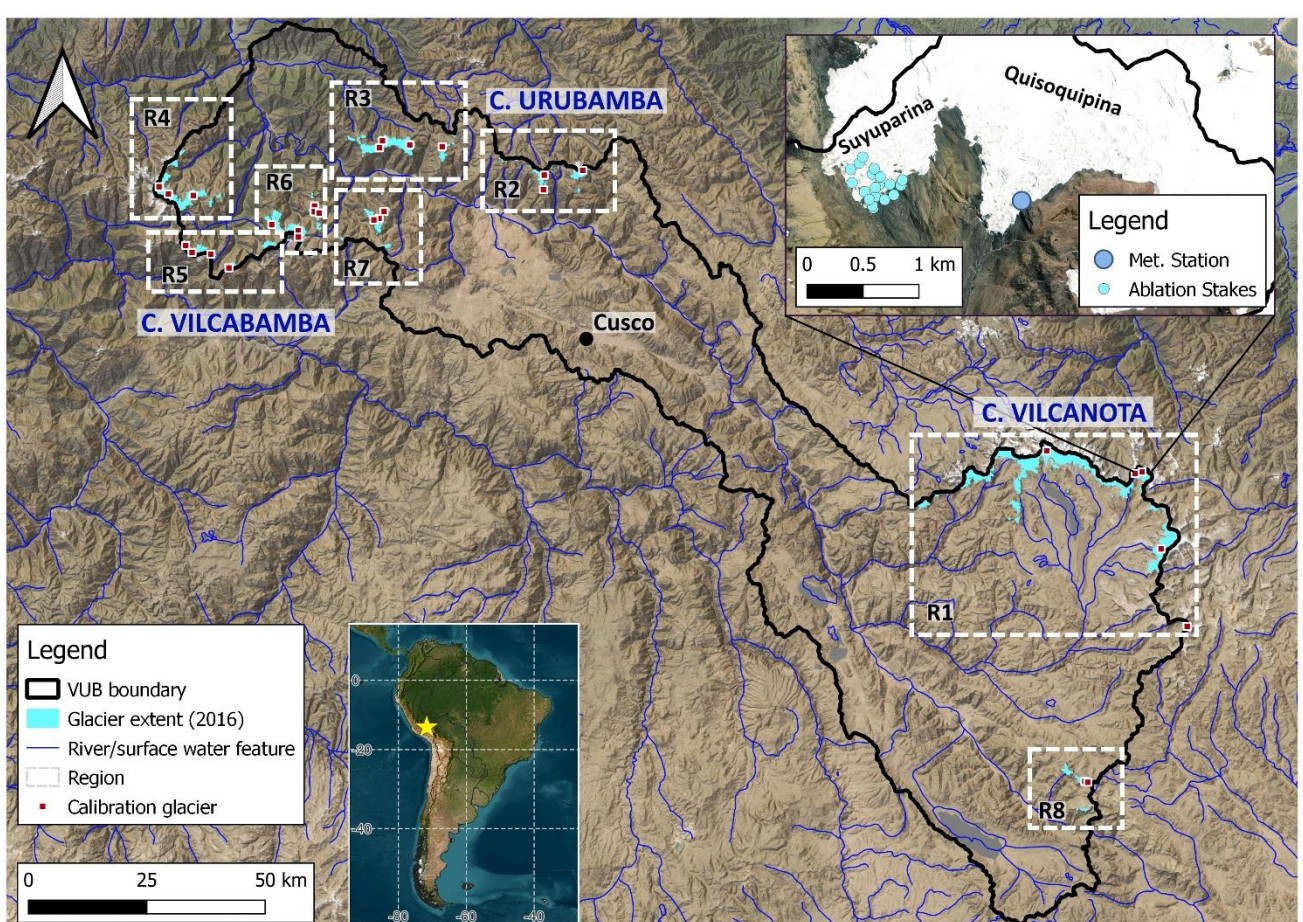

**Figure 1: Vilcanota-Urubamba basin. Inset bottom left shows location of basin in South America. Inset top right shows Quisoquipina and Suyuparina glaciers with meteorological station and ablation stake network. Satellite imagery sourced from Esri World imagery: Esri, Maxar, Earthstar Geographics, and the GIS User Community.**

## 2.2 Climate data

Hourly historical meteorological variables including near surface air temperature, wind speed, total precipitation, specific
humidity, surface air pressure and incident short and long wave radiation were taken from the Weather Research and
Forecasting (WRF) model, run from 1980 to 2018, with an outer domain with 12 km grid spacing and inner domain covering
the VUB with 4 km grid spacing. The precipitation and temperature variables were bias-corrected using station data from
within and around the VUB. Equivalent data for the other variables were not available and, as such, these were not bias-
adjusted. The bias-corrected temperature and precipitation outputs were used as the 'historical truth' to statistically downscale
30 global climate models from the Coupled Model Intercomparison Project 5 (CMIP5), to provide a 30-member ensemble of
"future" hourly temperature and precipitation driving data from 2019 to 2100 for both RCP4.5 and RCP8.5 emission scenarios.
The statistical method used was a variation on quantile mapping, following Cannon et al. (2015). This statistical method
preserves the trends in the original CMIP5 models, while adjusting the absolute magnitude of temperature and precipitation
and the number of wet days. While statistical downscaling methods like quantile mapping have been applied widely to
temperature and precipitation variables in the past, they are not routinely used for other variables such as wind and radiation
because of the weaker direct link between these variables across spatial scales. For example, a future increase in meridional
near-surface wind in the CMIP5 models would not necessarily translate into a future increase in near-surface wind at the WRF
model resolution due to the influence of local scale topography for example. Similarly, radiation change may depend more on
local temperature and humidity variability than large scale radiation change. An additional challenge of applying statistical
downscaling techniques to the other meteorological variables would have been the lack of validation data as the meteorological
stations in the VUB only provide temperature and precipitation variables. For these reasons, we decided not to attempt to
statistically downscale the CMIP5 projections of the other variables as this could have introduced additional unknown biases
in our input data. Instead, equivalent future data for the other meteorological variables were generated by resampling
(repeating) the 1980-2018 WRF simulations to produce a continuous 2019-2100 time series. By resampling in this way, we
preserve the year-by-year sequencing of the climate variables. This approach was deemed preferable to randomly resampling
the data as it preserves any multi-year cycling e.g. ENSO which may be present in the driving data. The data were manually
checked to ensure that this approach did not introduce any strange jumps in the driving data across the 2018 to 1980 crossover.
Full details of the WRF modelling setup, the bias-correction and the future statistical projections can be found in Potter et al.
(2023).

In addition to the gridded climate data, daily on-ice meteorological measurements between May 2012 to October 2016 were
made available after the work by Suarez et al. (2015). These were collected using an automatic weather station, which was
situated in the ablation zone of Quisoquipina glacier (5180 m asl) in the north-east of the VUB (Figure 1 inset top right). The
measurements include air temperature, relative humidity, incoming/outgoing shortwave and longwave radiation, wind speed
and wind direction.

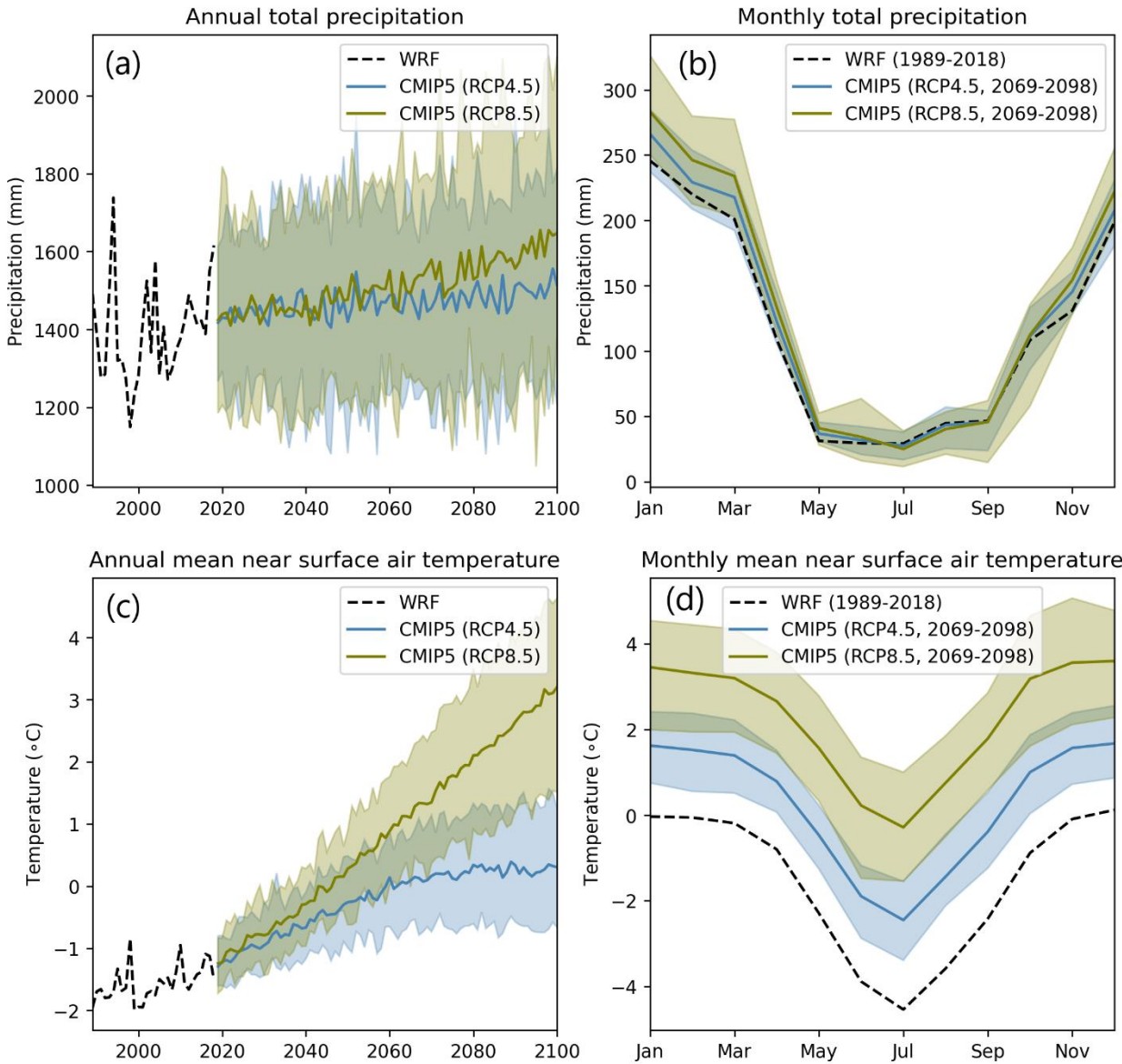

**Figure 2: Mean annual on-ice precipitation (a) and temperature (c) and monthly average on-ice precipitation (b) and temperature (d) based on a single WRF historical climate simulation and ensemble of CMIP5 simulations with 90% confidence bounds. Data taken from Potter et al. (2023).**

### 2.3 JULES-OGGM glacier modelling workflow

JULES-OGGM is a workflow for simulating glacier evolution with climate forcing using the physically-based energy balance and ice flow modelling schemes in JULES and OGGM respectively. It facilitates the exchange of data and feedbacks between these models. This section describes the approach used to integrate them into a single workflow. The reader is referred to Best et al. (2011) and Maussion et al. (2019) for a more detailed explanation of the models.

### 2.3.1 Climate data pre-processing

JULES requires continuous time series of meteorological forcing variables including near surface air temperature, incident long and shortwave radiation, air pressure, specific humidity and precipitation. The approach to deriving these is likely to be application specific. In the VUB, given glaciers are small ($< 8$ km$^2$) relative to WRF node areal coverage (16 km$^2$), hourly driving data for each glacier were extracted from the climate model node closest to the centroid of the glacier. Analysis of temperature, surface air pressure and specific humidity simulations from WRF, when taken from the model nodes in the

vicinity of the glaciers, showed lapse rates with a systematic seasonal cycle (Appendix A). Accordingly, 30-day moving average, hourly lapse rates were calculated for each glacier individually, estimated from the four nodes surrounding the glacier centroid. The median of these was then taken to estimate a regional lapse rate. This averaging step was necessary to remove, what were deemed to be unrealistically high or low lapse rates inferred at some glaciers. Analysis of downward shortwave radiation and windspeed indicated no clear lapse rate and so these were assumed constant with elevation. However, incident

shortwave was adjusted to account for the fact that the flux from WRF is for a flat horizontal surface whereas the glaciers have a tilted surface. The ratio of the direct incident shortwave radiation received on a tilted surface with respect to a horizontal surface can be calculated from the geometric factor, $R_b = \cos\theta / \cos\theta_z$, where $\theta$ is the angle of incidence and $\theta_z$ is the solar zenith angle. We calculated hourly $R_b$ over the simulation period using glacier slope and aspect derived from the SRTM digital elevation model and hourly simulations of the position of the sun in the sky which was calculated using the Pysolar python

package (version 0.11). The WRF incident shortwave radiation flux includes both direct and diffuse shortwave radiation. By default, JULES assumes that half of the incident shortwave radiation is direct and there was no clear justification to change this. Accordingly, the adjustment was only applied to half of the shortwave radiation flux provided by WRF. Downward longwave radiation is adjusted for elevation internally within JULES as a function of temperature following Shannon et al. (2019). Given the potential for significant bias in precipitation at high elevation due to the lack of observation data, the

precipitation lapse rate was reserved as a calibration parameter.

### 2.3.2 JULES

JULES resolves land surface processes including the surface energy balance. The model domain is discretised into one or more grid box nodes which can be further-discretised into tiles to incorporate sub-grid heterogeneity of the land surface and subsurface e.g. ice cover and soil properties respectively. Each grid box node resolves mass and energy fluxes in the vertical

direction only. Because of this, computational efficiency can be improved by specifying the model grid as a set of discontinuous grid box nodes where spatially-continuous outputs are not required. In the same vein, runtime may be improved by switching off process schemes entirely. Of relevance to glacier modelling is the multi-layer snowpack scheme. The scheme solves the full energy balance at the surface, given atmospheric forcing, following Eq. (1):

$$C_s \frac{\delta T_*}{\delta t} = (1 - \alpha)Sw_\downarrow + \epsilon Lw_\downarrow - \sigma\epsilon(T_*)^4 - H - L_c E - G, \tag{1}$$

where $C_s$ is the heat capacity of the surface (J m$^{-2}$ K$^{-1}$), $T_*$ is the surface temperature (K), $\alpha$ is the surface albedo, $Sw_\downarrow$ in the downward solar radiation at the surface (W m$^{-2}$), $\sigma$ is the Stefan Boltzmann constant (W m$^{-2}$ K$^{-4}$), $\epsilon$ is the surface emissivity, $Lw_\downarrow$ is the downward longwave radiation at the surface (W m$^{-2}$), $H$ is the turbulent heat flux (W m$^{-2}$), $L_c$ is the latent heat of condensation of water at 0ºC (J kg$^{-1}$), $E$ is the turbulent moisture flux (kg m$^{-2}$ s$^{-1}$) and $G$ is the heat flux to the material beneath (W m$^{-2}$).

The snowpack scheme simulates vertical heat transfer between snowpack layers, snow compaction, ageing (grain size and darkening) and age/density-dependant albedo evolution which is simulated using a prognostic albedo scheme based on the Wiscombe et al. (1980) spectral snow model. Rainfall on the snowpack percolates through the layers if the pore space is sufficiently large, while any excess water contributes to surface runoff. Liquid water below the melting temperature can refreeze. Shannon et al. (2019) configured a 0.5º JULES model to simulate global glacier volume projections for the 21$^{st}$ century. Each grid box was configured with 46 tiles with elevations ranging from 0 – 9000 m in increments of 250 m. For each grid box with glacier coverage, an initial dense ice layer at the base of the snowpack was included with proportional ice coverage across the tiles according to the cumulative observed hypsometry of glaciers within each grid box. Using hourly climate forcing data Shannon et al. (2019) simulated changes in glacier volume, but without any representation of ice flow.

In this study, version 6.0 of JULES was used and, following Shannon et al. (2019), a 10-layer snow and ice pack was used. The first nine layers include 5 m of snow and firn with depths of 0.05, 0.1, 0.15, 0.2, 0.25, 0.5, 0.75, 1, and 2 m. The bottom layer was used to represent ice with a given thickness of 500 m and initial density of 917 kg m$^{-3}$. The thickness was arbitrarily large to ensure that the entire ice thickness did not melt out at any JULES node during the simulation period. Given the thickness of the bottom layer and the fact that JULES does not account for layer density in its estimation of liquid water holding capacity, this capacity was set to zero to prevent excessive liquid water storage and refreezing. The snow and ice layer density, grain size, temperature and albedo were dynamically-initialised using 20 years (1981-2000) of hourly historical climate data. All model parameters, except those perturbed as part of the model calibration exercise (detailed below) were set to default values. All JULES modelling was performed on the UK NERC-JASMIN High-Performance Computing facility. A 120-year simulation for a single glacier (10 JULES grid boxes) takes ~25 minutes to run on a single processor. All 10 JULES grid boxes were used for every glacier to ensure that the glacier hypsometry, regardless of its evolution over the simulation period, was covered by the JULES grid box node elevations.

**2.4.2 OGGM**

OGGM simulates glacier surface mass balance and ice flow. For any glacier with geometric and climate forcing data, OGGM represents it as one or more connected flowlines which are discretised into nodes and parameterised with approximations of glacier width, area, thickness and bed shape. Ice flow is simulated by solving the continuity equation along each flowline:

$$\frac{\partial s}{\partial t} = w\dot{m} - \nabla \cdot uS, \tag{2}$$

where $S$ is the area of the cross-section perpendicular to the flow line, $w$ is the width of the cross-section, $\dot{m}$ is the mass balance [kg m$^{-2}$ s$^{-1}$] and $u$ is the average ice flow velocity (m s$^{-1}$), which includes ice deformation ($u_d$) and basal sliding ($u_s$). The ice deformation component is calculated using the shallow-ice approximation:

$$u_d = \frac{2A}{n+2} h\tau^n, \qquad\qquad\qquad\qquad\qquad\qquad\qquad\qquad (3)$$

where $A$ is the ice creep parameter (s$^{-1}$ Pa$^{-3}$), $h$ is the local ice thickness, $\tau$ is the basal shear stress and $n$ is the exponent in Glen's flow law. OGGM has an in-built temperature index model which runs alongside the ice flow model to simulate glacier evolution given temperature and precipitation forcing data.

OGGM version 1.4 with the default parameterisation for ice dynamics ($n = 3$, $A = 2.4\text{e-}24$ s$^{-1}$ Pa$^{-3}$ and $u_s = 0$ m s$^{-1}$) was used throughout this study. The initial glacier hypsometry and thickness was established using the in-built OGGM functionality based on the glacier outlines from Drenkhan et al. (2018) and the SRTM digital elevation model. Here, ice thickness is estimated at each flowline node by solving the steady state ice flow at each node. To do this, OGGM version 1.4 uses an "equilibrium mass balance" profile, where the cumulative mass balance across the glacier sums to zero. The thickness at each node is derived from the cumulative upstream surface mass balance. OGGM uses the in-built temperature index model to generate the equilibrium mass balance. For JULES-OGGM, this feature was bypassed and, instead, an equilibrium mass balance for each glacier was derived from the calibrated JULES model simulation over the historical period (1981-2018). To satisfy the equilibrium mass balance requirement, for each glacier, the mean mass balance at each glacier node was scaled to preserve the mass balance-elevation distribution whilst summing to zero. All OGGM modelling was performed on UKRI-BGS High-Performance Computing facility. A 100-year simulation for a single glacier takes ~90 seconds to run on a single processor.

### 2.4.3 Sequential coupling of JULES-OGGM

The basic principle of the JULES-OGGM workflow is to bypass the temperature index model in OGGM and drive the ice dynamics flowline model with surface mass balance simulations from JULES i.e., substituting $\dot{m}$ in Eq. 2 at each OGGM flowline node. Given the elevation-dependence of surface energy balance dynamics, JULES-OGGM must also account for influences of glacier geometry changes on surface mass balance. A workflow has been established that meets both of these requirements through two consecutive modelling steps that does not require dynamic model coupling:

**Step 1: Generating mass balance data for each glacier using JULES**

Each glacier is represented by $N$ ice-covered JULES grid boxes with elevations equally spaced between $z_{min}$ and $z_{max}$. For this study, $z_{min}$ and $z_{max}$ were set to 4000 and 6500 m asl respectively over $N=10$ grid box nodes equally spaced by ~278 m elevation. These elevations do not change during a simulation. Rather, the elevation range was selected to bound the elevation of all glacier ice over the simulation period while $N$ was selected to provide adequate representation of changes in mass balance with elevation. Climate data are pre-processed for each grid box using the prescribed lapse rates to represent the climate at that elevation for each glacier. JULES is configured to output annual accumulated snow and ice mass change (specific mass

balance) at each grid box elevation of each glacier (hypothetical data for single glacier given in Figure 3a). The outputs from JULES, therefore, indicate how surface mass balance changes as a function of elevation only on each glacier. No other factors
affecting the spatial variability in mass fluxes are accounted for at this stage.

**Step 2: Driving OGGM with mass balance outputs from JULES**

To bypass OGGM's built-in temperature index model, the OGGM source code (version 1.4) was modified to include a new function which reads in the mass balance outputs from JULES. For a given glacier, OGGM takes the annual specific mass balance from the JULES simulation over the $N$ grid boxes (Figure 3b). For each OGGM node of that glacier, the mass balance
at the OGGM node elevation is extracted from the JULES grid box simulations (yellow dash arrows linking, Figure 3b and d), using linear interpolation. This approach allows for incrementation of the glacier mass balance with elevation and also implicitly accounts for the elevation feedback to surface mass balance without requiring two-way coupling between JULES and OGGM. The elevation feedback is demonstrated in  Figure 3c, where the subsequent retreat of the glacier over time results in the lowering of the ice surface (Figure 3e), and consequently, OGGM extracts the mass balance from the JULES simulations
at lower elevations. OGGM is configured to output annual changes in glacier volume and area.

One important consideration in the presented JULES-OGGM coupling is how the model deals with the varying physics of surface energy fluxes on ice-free and ice-covered regions as a glacier advances and retreats in OGGM. This could be important, particularly when a glacier is advancing, as OGGM considers ice flowing into ice free nodes in combination with surface accumulation and ablation processes when calculating the redistribution of ice mass into previously ice-free regions. It was
decided to ignore these differences, in part because including an ice-free representation in JULES would effectively double the computational costs, but also because all evidence suggests that glaciers in the VUB are and will continue to be in a retreat phase over the planned simulation period which would render any differences in ice-free and ice-covered physics inconsequential. The implications of this simplification for this study and for other studies will be explored further in the discussion.

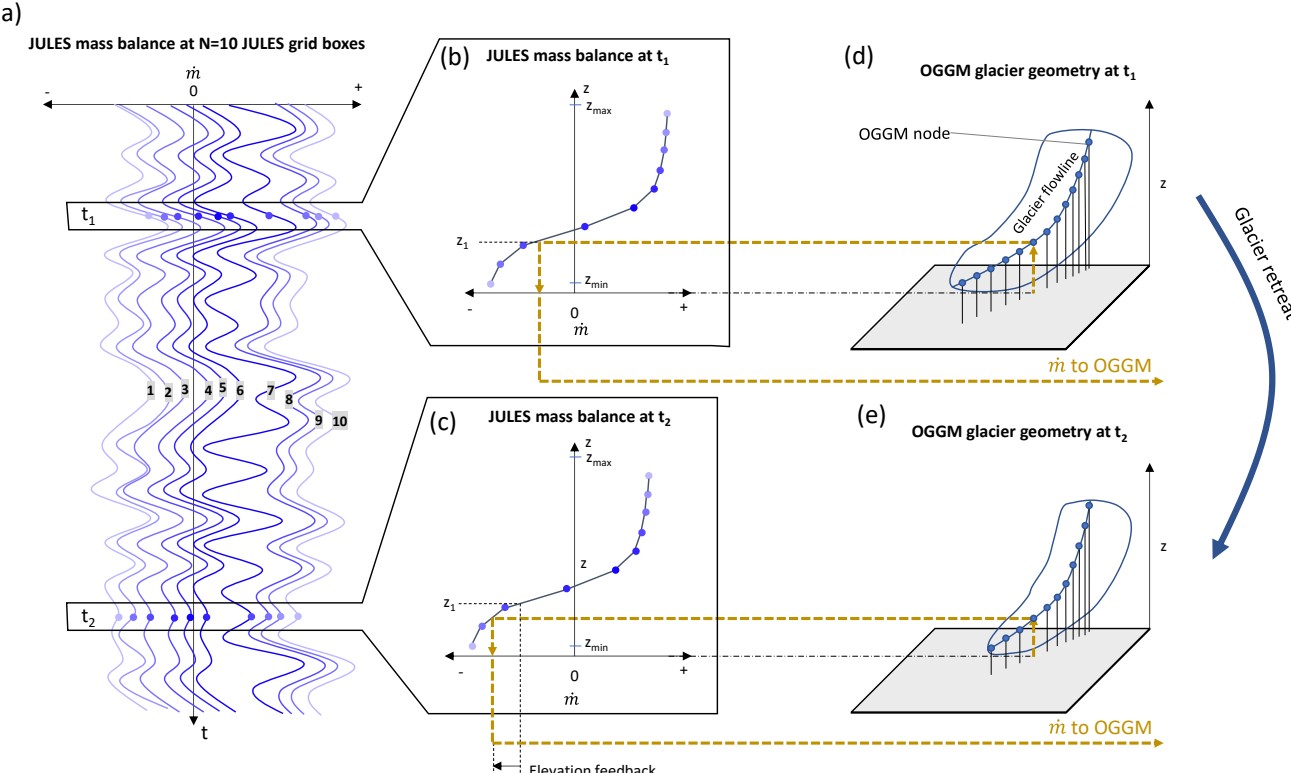


**Figure 3: Hypothetical application of JULES-OGGM to single glacier including annual simulated specific mass balance at N=10 JULES grid boxes at different elevations on the glacier (a); the simulated annual mass balance at the 10 grid boxes for time=t₁ (b); and time=t₂ (c); and the glacier flowline representation in OGGM at t₁ (d); and t₂ (e). The yellow dash lines represent the mass balance extraction process from JULES to OGGM.**

**2.5 Model calibration, evaluation and sensitivity analysis**

JULES and OGGM both use model parameters that are likely to be spatially variable, and cannot easily be constrained from observation data. In this study, only the parameters of the energy balance in JULES were considered for calibration, while the default parameters for the ice flow component of OGGM were used. An iterative calibration strategy was undertaken to tune the JULES parameters to achieve the best fit to the geodetic mass balance data. Firstly, manual perturbations of the JULES

model parameters were undertaken to identify those that exhibited sensitivity. We began with all seven parameters identified by Shannon et al. (2019) except for the temperature lapse rate. These include the fresh snow and ice albedo (for visible and near infrared wavelengths), the precipitation gradient and a wind speed scale factor. We also included: i) the roughness length of momentum given its potential importance for sublimation processes; ii) the temperature threshold below which precipitation falls as snow; iii) the density above which snow is considered to be firn; and iv) the weighting of albedo between visible and

near-infrared wavelengths. From these experiments, we identified four potentially-important parameters for model calibration and suitable calibration ranges (Table 1).

**Table 1: Calibration parameters and ranges.**

| Parameter | Description | Calibration range |
|---|---|---|
| $z_0$ | Roughness length of momentum | 1 - 100 mm |
| aicemax | Maximum albedo of bare ice | 0.2 - 0.6 |
| wght_alb | Weighting of albedo between visible and near-infrared wavelengths | 0.5 - 0.9 |
| $\gamma_{precip}$ | Orographic precipitation gradient | 0-10% /100 m |

Having established the calibration parameters, model calibration was undertaken using the concept of Monte Carlo whereby random perturbations of these parameters were tested and assessed for their goodness-of-fit to glacier-wide estimates of specific mass balance over the 2000-2018 period from the geodetic observation data. The aim of the calibration procedure was to select the parameter set for JULES that minimised the sum of the glacier-area-weighted root mean squared error ($RMSE_w$). A glacier-area-weighted bias (mean error), was also calculated ($BIAS_w$). Computational constraints limit the number of glaciers

and number of random perturbations which can reasonably be implemented in the framework. Accordingly, a subset of 30 glaciers were identified for the calibration procedure (red squares, Figure 1), selected to include glaciers with a range of elevation, aspect and slope characteristics and with good spread across the study basin. Given the uncertainty associated with geodetic measurements of small glaciers, those with the largest overall areas were selected where possible. 1000 random perturbations of the parameters were tested and assessed for their goodness-of-fit to the geodetic data. The quasi-random Sobol

sampling strategy (Bratley et al., 1988) was used to sample the parameter space efficiently. All parameters from sampled from a uniform distribution.

The calibration parameters were assessed for sensitivity using the PAWN density-based global sensitivity analysis method (Pianosi et al., 2015) which uses cumulative distribution functions from model outputs conditioned on different parameterisations to estimate sensitivity to different parameters. The PAWN method provides a quantitative, global sensitivity

analysis that has been used across environmental modelling applications (Amaranto et al., 2020; Laarabi et al., 2022; Lin et al., 2020). It can be applied to outputs from generic Monte-Carlo experiments (Pianosi et al., 2018) and uses the Kolmogorov–Smirnov statistic as the basis for quantifying parameter sensitivity making it easy to interpret and compute. In this study, we calculated the PAWN sensitivity for each parameter across each calibration using the $RMSE_w$ scores from the Monte-Carlo experiment. All analysis was undertaken using the safepython python library version 0.2.0 (Pianosi et al., 2015). 95%

confidence intervals were estimated for all metrics using the in-built bootstrapping algorithm. Following Pianosi et al. (2018), a dummy parameter, which has no impact on the model outputs, was introduced to the analysis to estimate the magnitude of approximation errors.

The calibrated parameters were then incorporated into the basin-wide JULES-OGGM model which was subsequently evaluated for robustness against the geodetic observations of specific mass balance over all 532 glaciers in the VUB as well

as against the glaciological mass balance observations on Quisoquipina and Suyuparina glaciers.

# 3 Results

## 3.1 Model calibration and sensitivity analysis

As a first assessment of the model appropriateness, the optimal simulation from the Monte Carlo experiment was extracted for each glacier individually. Figure 4a shows that, for all calibration glaciers, it is possible to achieve a close fit to the observed specific mass balance when the calibration parameters are tuned to each glacier separately. JULES-OGGM is able to capture the observed specific mass balance within 0.008 m w.e. of the observations for all glaciers. When the single best parameterisation is chosen based on the $RMSE_w$ across all glaciers, the errors are larger (Figure 4b) ranging between -0.99 and 1.22 m w.e. with a $RMSE_w$ of 0.56 m w.e. It would not be feasible to tune the parameters to each glacier individually. However, a single parameterisation appears to degrade the performance of the model significantly. Accordingly, it was decided to investigate if an improved goodness-of-fit could be obtained by adopting a regional-parameterisation approach. Here, we split the calibration glaciers into eight different regions (dashed white boxes in Figure 1), in part reflecting known differences in glacier hypsometry and driving climate characteristics, and to ensure that at least two glaciers are obtained within each region. This resulted in eight parameterisations which resulted in a modest improvement of the $RMSE_w$ to 0.46 m w.e. (Figure 4c). While the errors remain significant, the errors were reduced for 21 of the 30 calibration glaciers, and the spatial disaggregation allows for ease of upscaling the parameterisation to all glaciers in the catchment and so, this regional approach was adopted for the remainder of the study. Note that, because of this, regional lapse rates were also applied (Appendix A).

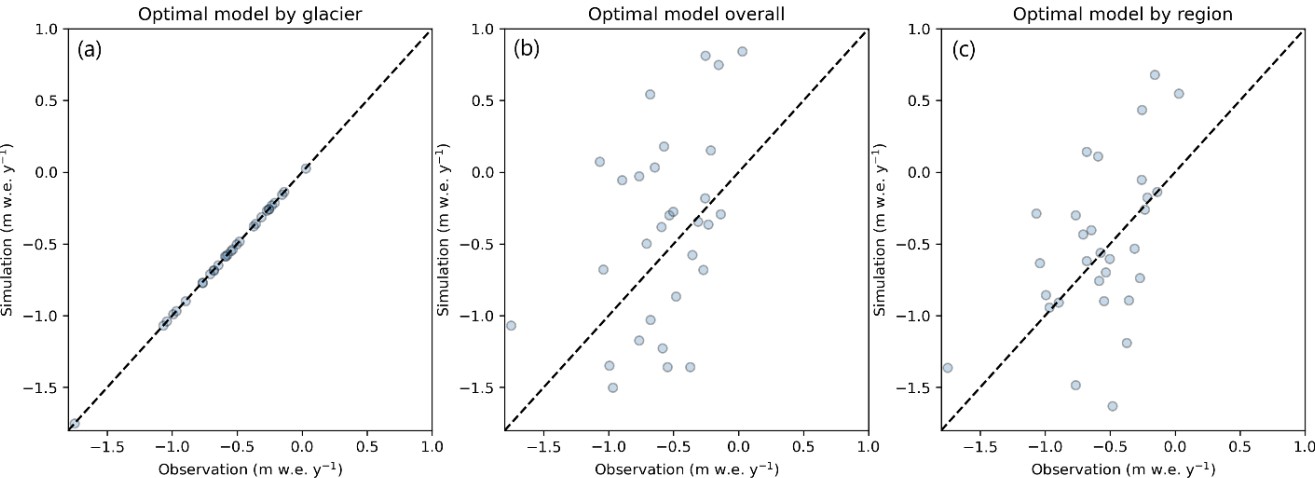

**Figure 4: Simulated and observed mean annual specific mass balance over 2000-2018 for the 30 calibration glaciers using different calibrated parameterisations including those obtained for each glacier separately (a); when using the single best overall parameterisation based on the $RSME_w$ score (b); when using the models optimised for the eight delineated regions (c).**

The RMSE$_w$ and BIAS$_w$ scores range between 0.01-0.73 m y$^{-1}$ w.e. and -0.06-0.29 m y$^{-1}$ w.e. respectively across the regions (Table 2). The calibrated parameters typically span a large portion of the calibration range. The only exceptions to this is $z_0$

which does not exceed 7 mm for any of the regions.

**Table 2: Calibrated parameter sets from the calibration parameters in Table 1 for each region and RMSE$_w$ and BIAS$_w$ scores.**

| Region | $z_0$ (mm) | aicemax | wght_alb | $\gamma_{precip}$ (%/100 m) | RMSE$_w$ (m y$^{-1}$ w.e.) | BIAS$_w$ (m y$^{-1}$ w.e.) |
|--------|-----|---------|----------|------|------|-------|
| 1 | 5.01 | 0.26 | 0.77 | 9.48 | 0.49 | 0.08 |
| 2 | 2.82 | 0.25 | 0.62 | 1.60 | 0.01 | -0.002 |
| 3 | 1.17 | 0.59 | 0.58 | 9.45 | 0.38 | -0.001 |
| 4 | 1.62 | 0.25 | 0.64 | 2.07 | 0.49 | 0.15 |
| 5 | 1.49 | 0.50 | 0.71 | 5.26 | 0.21 | -0.06 |
| 6 | 1.52 | 0.42 | 0.67 | 0.86 | 0.47 | 0.11 |
| 7 | 6.69 | 0.54 | 0.67 | 0.27 | 0.73 | 0.29 |
| 8 | 5.88 | 0.59 | 0.75 | 7.81 | 0.03 | -0.002 |

The PAWN sensitivity analysis reveals that for all of the regions, the modelled mass balance is most sensitive to the wght_alb

parameter (Figure 5). Furthermore, for all regions except for R6, the wght_alb parameter is the only parameter with a sensitivity index significantly greater than the magnitude of approximation errors, as shown by the dummy parameter. For R6, the simulated mass balance is also sensitive to the precipitation gradient, $\gamma_{precip}$. The simulated mass balance shows negligible sensitivity to the aicemax and $z_0$ parameters for all regions.

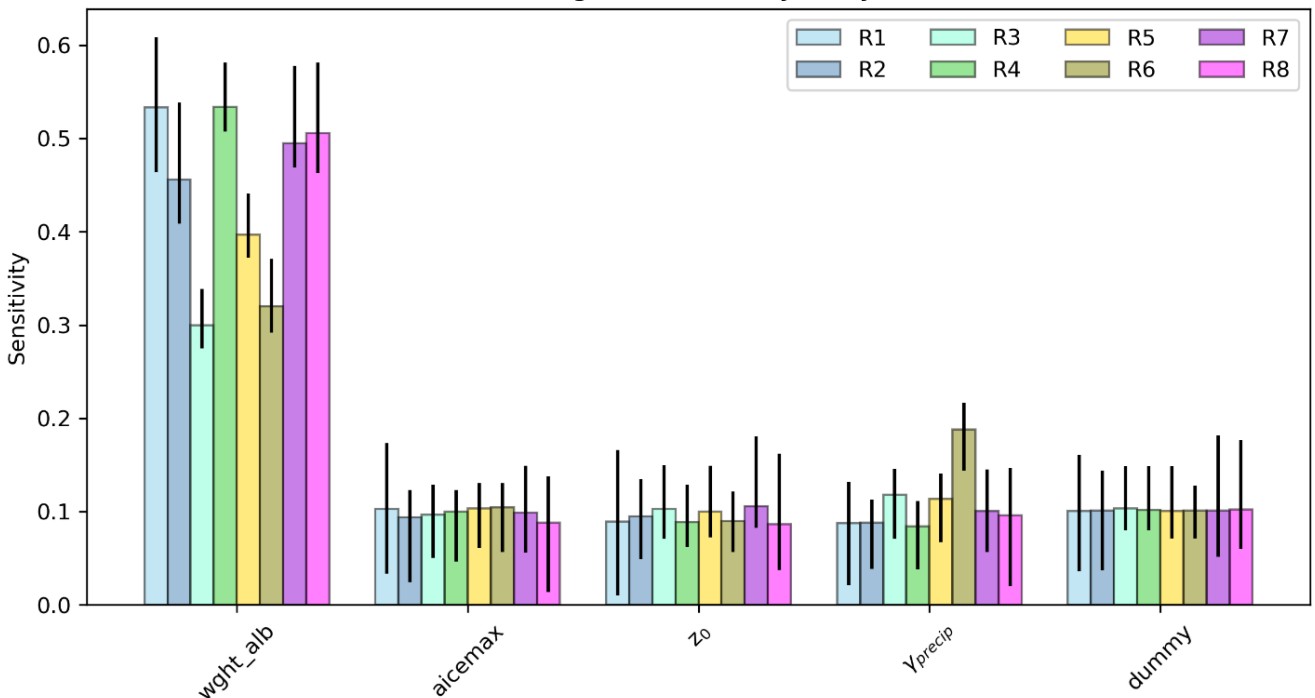

**Figure 5: Region-specific PAWN sensitivity metrics and 95% confidence intervals (black lines) for all calibration parameters and dummy parameter.**

## 3.2 Model evaluation

### 3.2.1 Basin-wide mass balance

When applied to all 532 glaciers, the simulated specific mass balance varies considerably, ranging from -4 to +4 m y$^{-1}$ w.e. which is in line with the observed range (Figure 6a). The area-weighted specific mass balance across all glaciers is -1.06 m y$^{-1}$ w.e., approximately double that derived from the geodetic observations (-0.52 m y$^{-1}$ w.e.), indicating that the model is losing mass too quickly over this period. In addition, the model does not capture within-basin variability in specific mass balance, with absolute errors as high as 3 m y$^{-1}$ w.e. The sources of these errors are not clear. When aggregated to 0.05 degree tiles,

there is considerable variability in the magnitude and direction of error across the basin and within the regions (Figure 6b). A comparison of the mass balance errors to glacier area, aspect, slope and elevation attributes showed no clear pattern of coherence to glacier characteristics (Figure B 1).

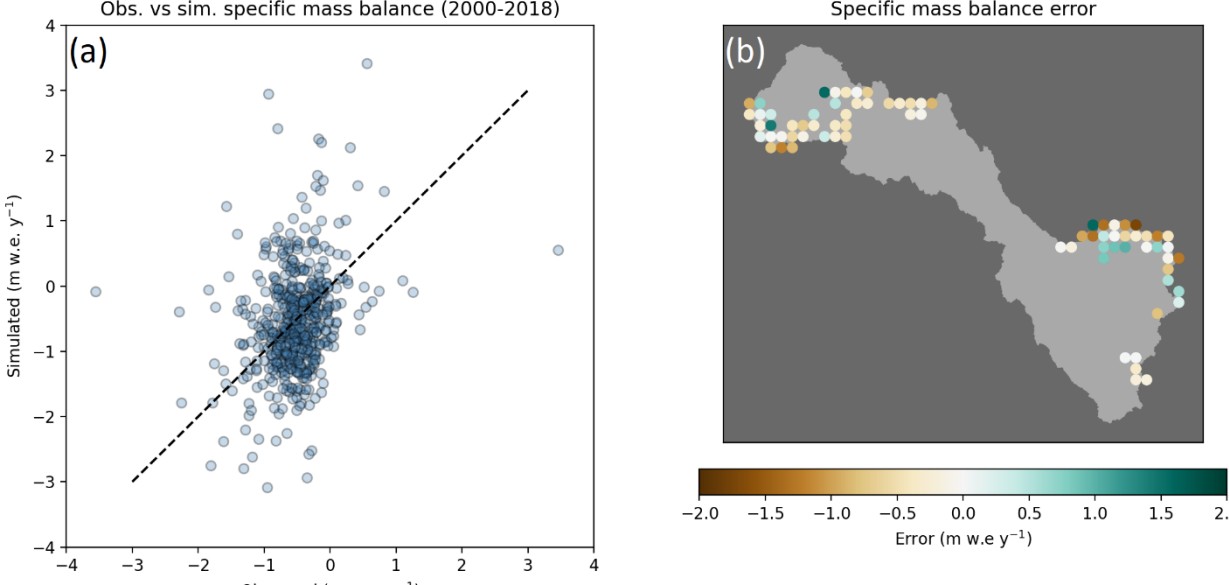

**Figure 6: Observed vs simulated annual mean specific mass balance (2000-2018) for all glaciers in the VUB (a) and area-weighted mean annual mean specific mass balance errors aggregated to 0.05 degree tiles (b).**

### 3.2.2 Glacier volume simulations

A comparison of the estimated glacier volumes from Millan et al. (2022) for the year 2018 against the JULES-OGGM simulation shows that there is generally good agreement, especially for the largest glaciers in the catchment (Figure 7). JULES-OGGM does, however, show a tendency to underestimate the volume of the smallest glaciers ($<10^{-2}$ km$^3$). The total JULES-OGGM ice volume for glaciers where Millan data are available is 7.19 km3 whereas Millan et al. (2022) estimate it as 8.16 km3.

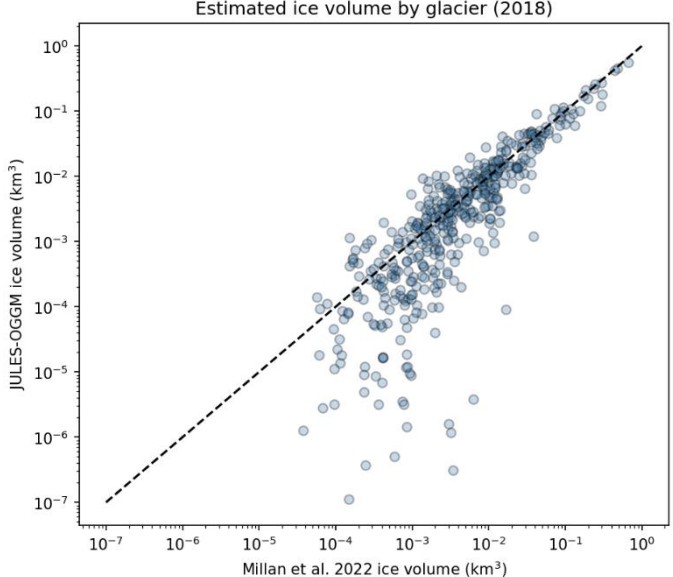

**Figure 7: Estimated glacier volumes from Millan et al. (2022) vs the JULES-OGGM ice volume simulation for 2018 for all glaciers where data are available in the VUB.**

### 3.2.3 Quisoquipina and Suyuparina glaciers

To aid the evaluation of the robustness of the calibrated model parameters, a point scale JULES model (no ice dynamics) was set up to evaluate against the ablation stake data on Suyuparina glacier. The model was parameterised with the corresponding R1 region parameterisation. All WRF driving climate variables were bias-corrected in the long-term mean using the daily Quisoquipina meteorological station data. Suyuparina and Quisoquipina glaciers are two of the 30 calibration glaciers. Bias-correction of the hourly WRF data, rather than using the raw daily meteorological station observations was chosen due to the requirement of hourly data to drive JULES and to ensure consistency with the calibration driving data. The model was run over an evaluation period of June 2012 to January 2016 where meteorological and ablation data are available.

Using this model, the energy balance simulations (Figure 8a) show that solar radiation ($SW_{net}$) is the dominant energy input for the majority of the simulation period, followed by sensible heat warming from the air ($H$). Latent heat fluxes ($L_cE$) are typically negative and smaller in magnitude. The net diffuse radiation flux ($LW_{net}$) has a marked seasonality: positive during the wet season (austral summer) when cloud cover is highest and negative during the cooler, clear sky days during the austral winter. Interestingly, $SW_{net}$ shows only weak seasonality, with a peak during the wet season. This appears to be related to the behaviour of albedo at the glacier surface (blue line, Figure 8c) which rarely falls below 0.8 and appears to be highest in the dry season. This is in contrast to the observed albedo (black line, Figure 8c) which typically peaks during the wet season when there is plentiful fresh snow and falls as low as 0.2 by the end of the dry season when fresh snow cover is minimal and the ice surface has darkened. Indeed, the simulated snow/ice layer density of the top 2 m (Figure 8e) shows that, even during the dry

season the top 5-10 cm typically resides at $< 500$ kg m$^{-3}$, representative of old snow. In reality, at the Quisoquipina meteorological station, bare ice is exposed seasonally. The inability to capture ice exposure and darkening through the dry season is consistent with the simulated cumulative ablation (yellow line, Figure 8g) underestimating observed ablation (black line, Figure 8g) in 2014. This period is highlighted between dashed lines in Figure 8g.

As an additional experiment, the observed daily proportion of outgoing to incident solar and diffuse radiation at Quisoquipina met station were used to convert the hourly WRF radiation inputs to net radiation: effectively bypassing the prognostic albedo model in JULES. When driving the model in this way it can be seen that the simulated $SW_{net}$ and surface albedo are much more seasonally-variable (Figure 8b and d respectively) and the simulated albedo dynamics are more coherent with the observations. The model also better-replicates the seasonal loss of the snow pack (Figure 8f) where the top 5-10 cm routinely exceed a density of 700 kg m$^{-3}$. What's more, within the period bounded by the dashed lines in Figure 8h, the simulated and observed ablation rates (gradients of yellow and black lines respectively, Figure 8h) are more closely-matched. Note, however, that the negative bias over the preceding wet season persists.

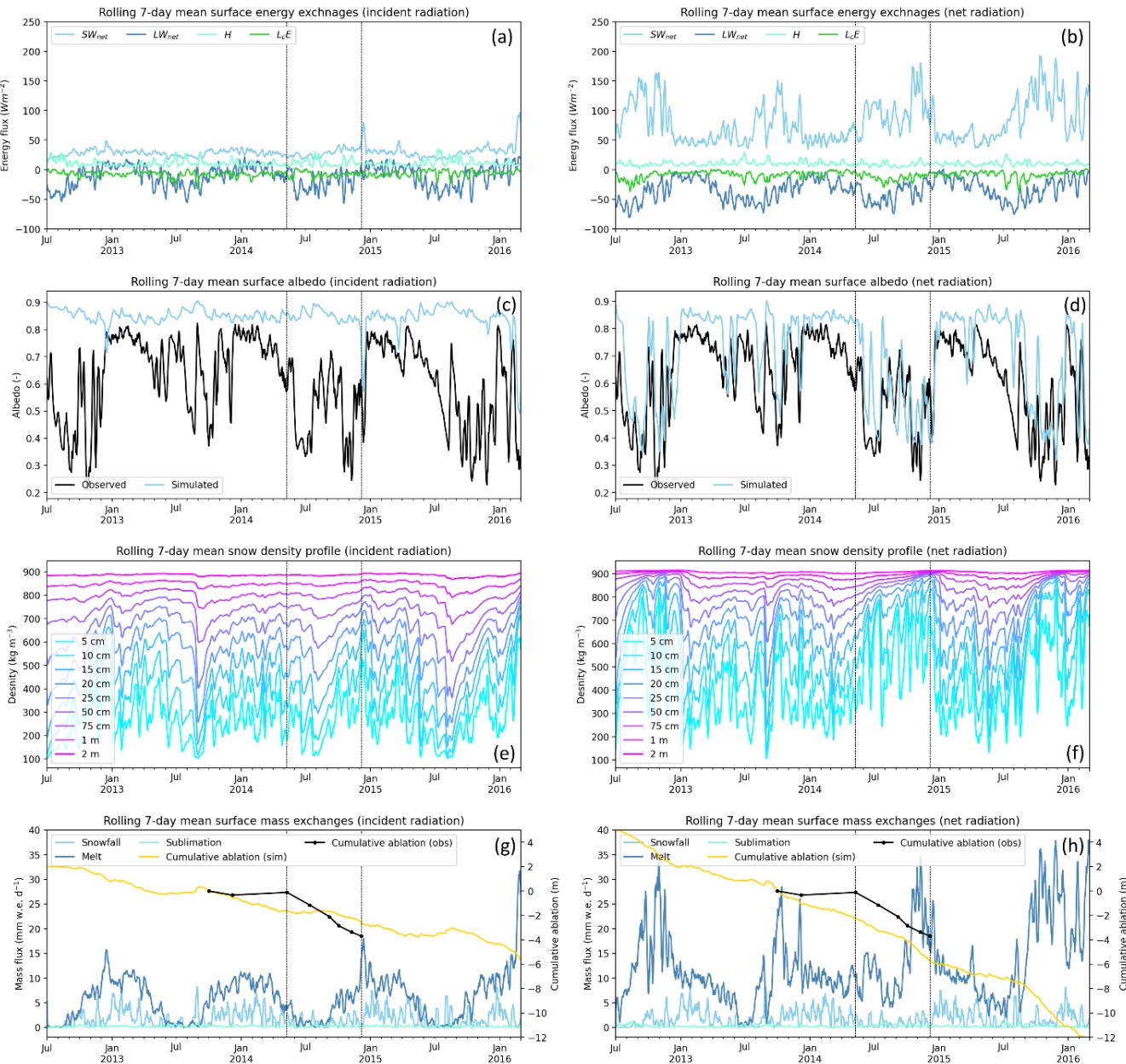

**Figure 8: JULES energy balance simulations on Quisoquipina glacier from July 2012 to January 2016 (when local meteorological and mass balance data are available) for model driven with incident radiation observations (a,c,e,g) and net radiation observations (b,d,f,h). For ease of interpretation, all variables are shown as 7-day moving averages. Note that sublimation, as output by JULES, is the net effect of sublimation minus deposition processes.**

### 3.3 Twenty-first century glacier projections

#### 3.3.1 Area and mass evolution

The projections indicate that total glacier area and mass in the VUB have decreased year-on-year from the year 2000 and will
continue to do so to the end of the 21$^{st}$ century under the RCP4.5 and RCP8.5 scenarios (Figure 9). Relative to 2000, the glacier
area is predicted to halve by 2047 (2043) under RCP4.5 (RCP8.5). The glacier mass is predicted to halve by 2027 for both
RCPs. This close agreement is likely reflective of the relatively similar temperature and precipitation signal across both RCPs
for the early-21$^{st}$ century. By the end of the century, the differences in predictions across the RCPs is more pronounced with
the total glacier area predicted to decrease to 28% (11%) of that in 2000 and the total glacier mass predicted to decrease to
17% (6%) of that in 2000 for RCP4.5 (RCP8.5).

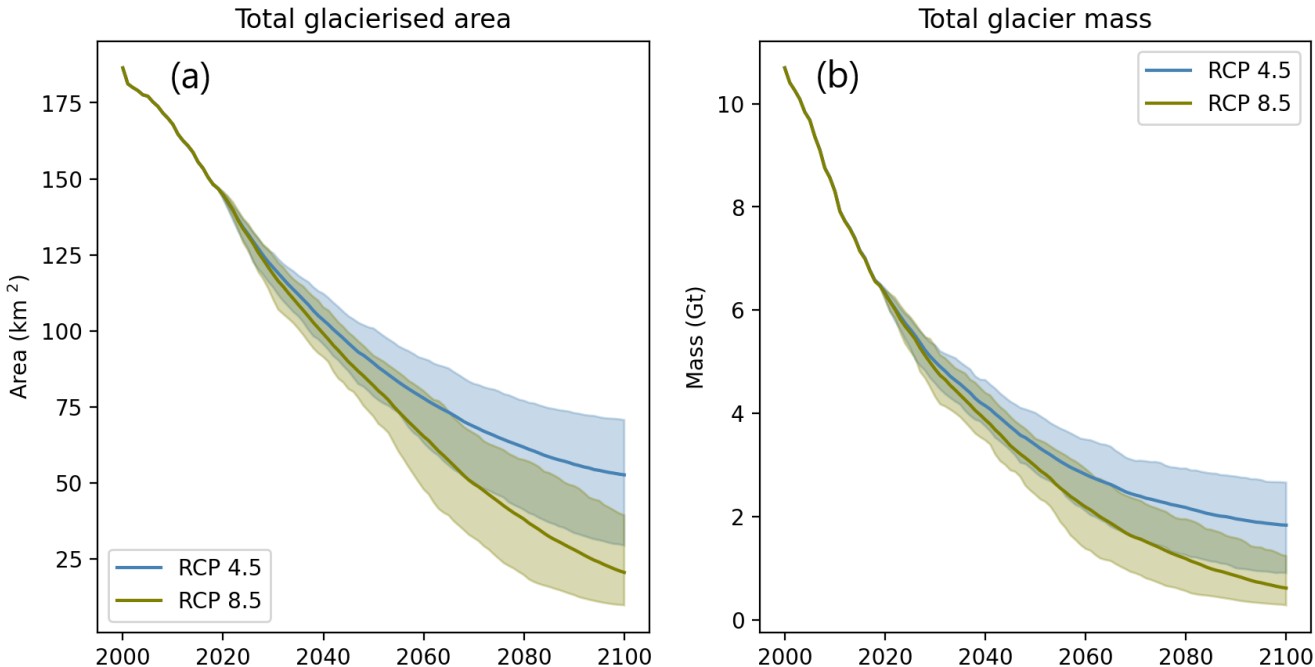

**Figure 9: Simulated annual total glacierised area (a) and mass (b) for the VUB between 2000 and 2100 based on RCP4.5 and RCP8.5
scenarios. Each RCP is represented by the ensemble mean (solid line) and 90% confidence interval (shaded area).**

#### 3.3.2 Variability in glacier response to climate change

K-means clustering was used to cluster the glaciers based on their ensemble mean simulated annual glacier mass time series
over the future period of 2020 to 2100 under RCP8.5. Four clusters were found to have distinct mass evolution dynamics
(Figure 10a and b). Clusters 1 to 3 all show sustained mass loss over the simulation period while cluster 4 shows initial mass
gain up to the middle of the century, followed by mass loss for the remainder of the century. The timing of this inflection in
cluster 4 is approximately 5 years earlier in RCP8.5. The mass loss rate is greater in RCP8.5 than RCP4.5 for all clusters.

From clusters 1 to 4, the proportional rate of retreat is progressively smaller. This corresponds to a progressively higher median elevation (Figure 10f). At the two extremes, clusters 1 and 4 are typically small in surface area and at low and high elevations respectively (Figure 10e and f). Cluster 3 accounts for the largest proportion (> 80%) of glacier mass for the majority of simulation period (Figure 10c). This type of glacier is the dominant in the Cordillera Vilcanota in the east of the catchment. The lower-elevation glaciers (clusters 1 and 2) are more prevalent to the west of the catchment.

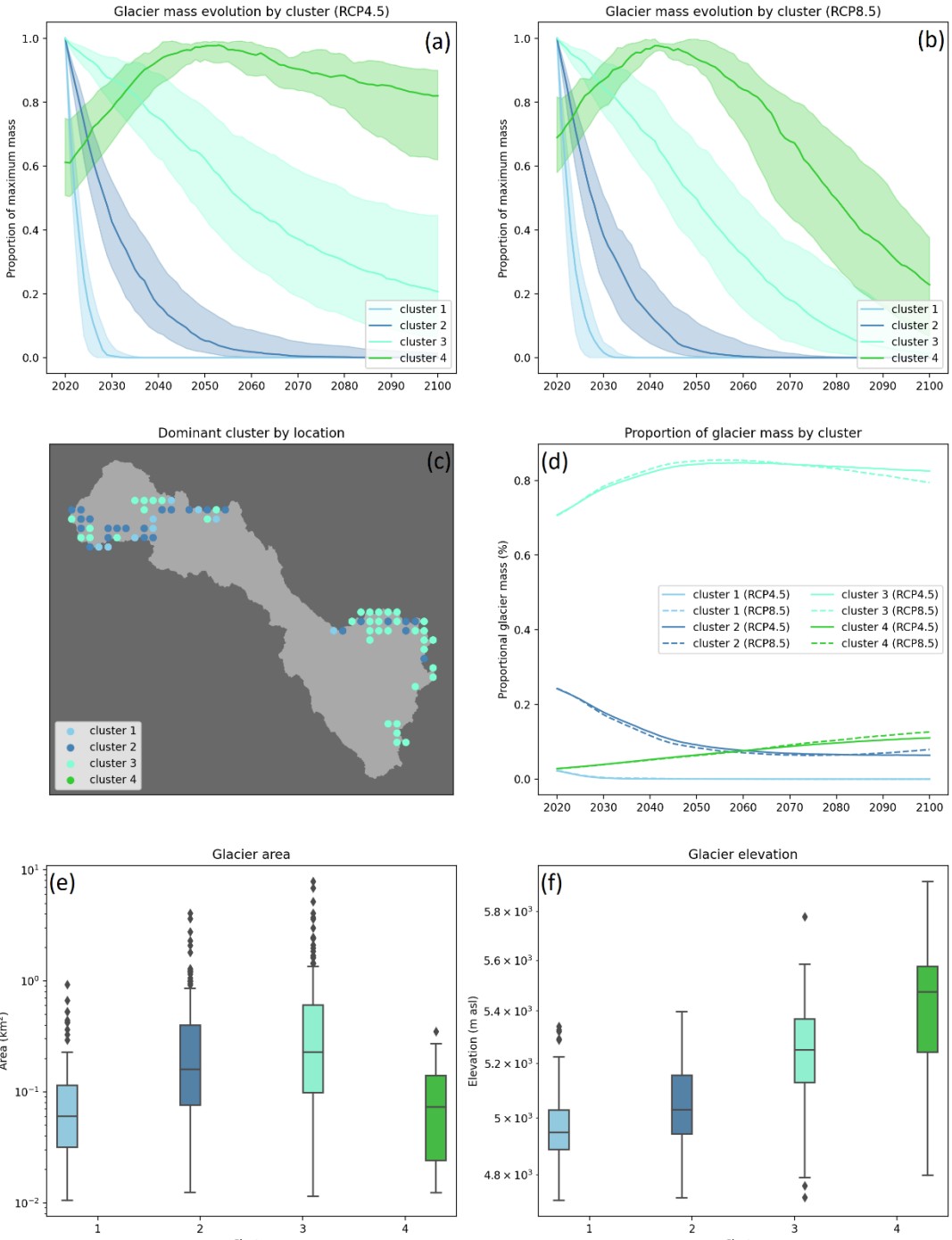

**Figure 10:** Annual glacier mass evolution of four identified clusters expressed as the proportion of the maximum simulated mass between 2020 and 2100 for RCP4.5 (a) and RCP8.5 (b) taken from the ensemble mean. Each cluster is represented by the median (solid line) and interquartile range (shaded area). Dominant cluster by 0.05 degree cell (c); evolution of glacier mass distribution

 **between clusters (d); spread of glacier area based on initial mapped areas for the year 1998 (e) and initial elevations based on SRTM elevation data for the year 2000 (f) within clusters expressed as box plots.**

### 3.3.3 Mass and energy balance dynamics

Annual changes in ice elevation, specific mass balance and energy balance have been calculated for each cluster and RCP combination (RCP8.5 in Figure 11 and RCP4.5 in Figure E 1). These variables have been calculated as an average across the active ice area only so as to avoid zero fluxes due to the complete loss of glacier ice. This raises an additional challenge when interpreting the results over an ensemble of climate inputs given that the distribution of glaciers will not be the same across all ensemble members over time. Simply averaging across the ensemble, therefore, has the potential to obscure co-dependencies between different mass and energy fluxes. Rather than taking the mean over the full ensemble of simulations, we have used a single global climate model simulation (BNU-ESM) from the CMIP5 ensemble which was found to most-closely follow the ensemble-mean mass evolution of the clusters under both RCPs (Figure D 1).

Mass balance (middle row in Figure 11 and Figure E 1) is consistently negative for all clusters except cluster 4 where we see a transition from positive to negative mass balance in the mid to late century. The remaining clusters show a positive trend in mass balance under RCP4.5, but a more mixed response under RCP8.5. These decadal trends are primarily controlled by changes in snow and ice melt. The melt flux is typically more variable over the 21$^{st}$ century than the snowfall input and much higher in magnitude than the net mass flux due to sublimation and deposition which is only very-weakly positive for all of the clusters under both RCPs. Snow and ice melt shows some coherence with changes in near surface air temperature: itself a function of climate warming and the retreat of glaciers up to cooler and higher elevations, a trend that is consistent across the catchment (top row in Figure 11 and Figure E 1). The apparent stability in the snowfall input appears to be related to a compensatory effect, where the warming and transition of snowfall to rainfall is counter-balanced by: i) the retreat of glaciers to higher elevations where precipitation rates are higher, and a smaller fraction of that precipitation falls as rain; and ii) the increase in total precipitation over time projected by the CMIP5 models. This, however, is only true for the larger glaciers of clusters 2 and 3 which span a large elevation range. For the smaller high-elevation glaciers (cluster 4), the reduction in mass input from snowfall due to rising temperatures is comparable to the magnitude of change in melt and serves to accelerate the retreat of the glaciers.

While there is generally good coherence between the snow and ice melt flux and changes in near surface air temperature, this co-variability does not always hold. For example, near surface air temperature over the glaciers in clusters 2 and 3 show a steady increase from 2040 onwards on average under RCP 8.5 (Figure 11). The average melt flux increases steadily at first, but then more rapidly from 2080 onwards, which is not reflected in the temperature data. Analysis of the surface energy balance components (bottom row in Figure 11 and Figure E 1), reveals that the melt flux is more coherent with changes in net shortwave radiation: the dominant source of energy at the snow/ice surface for all clusters under both RCPs which is itself, inversely proportional to the surface albedo. While higher temperatures facilitate more melt and inhibit snowfall, thereby lowering the albedo, it is changes in net shortwave radiation that primarily controls decadal variability in melt rates and surface mass

balance. Turbulent heat fluxes are smaller than radiative energy fluxes for all clusters and RCPs. Sensible heat fluxes typically increase with time, presumably due to the rising air temperature, while latent heat fluxes decrease.

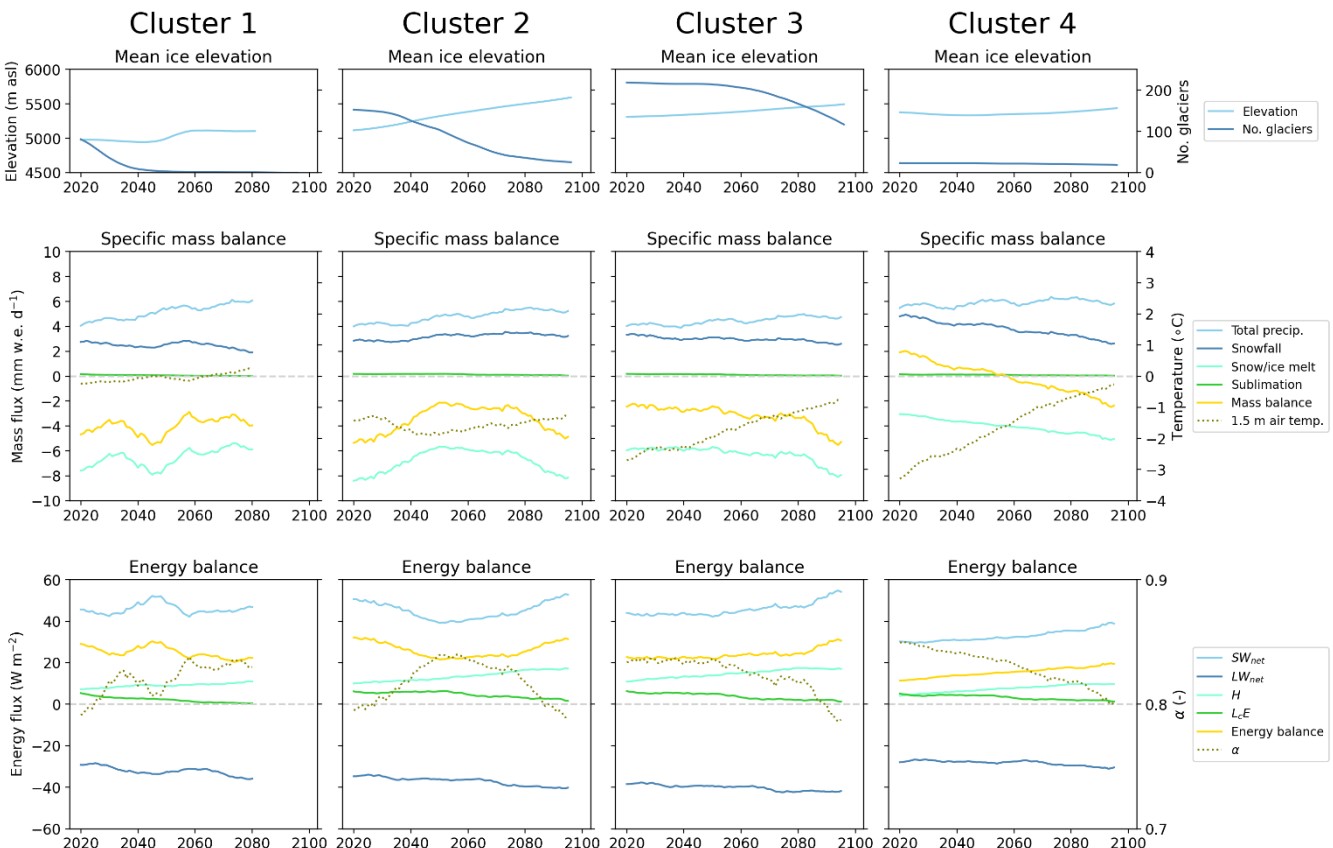

**Figure 11: Simulated mean glacier-wide state-variables from JULES-OGGM between 2020 and 2100 for each cluster (columns) and based on the BNU-ESM simulation under RCP8.5. The variables include ice elevation (top row), specific mass balance (middle row) and energy balance (bottom row). All variables are calculated over a 10-year moving average window and are taken as the mean across all glaciers within a cluster, weighted by the glacier area.**

## 4 Discussion

### 4.1 Model performance and limitations

The model calibration experiments suggest that the energy balance parameters in JULES-OGGM can be tuned to capture the decadal (2000-2018) glacier-wide mass balance of a range of glaciers in a tropical setting with a small error ($< \pm 0.01$ m y$^{-1}$ w.e.). This is encouraging for the future use of physically-based models for modelling tropical glacier evolution. However, attempts to extrapolate these parameters across all 532 glaciers in the VUB resulted in inconsistencies in model performance.

This extrapolation step was required given that the Monte-Carlo calibration approach was too computationally-demanding to afford tuning the model parameters to all glaciers individually like, for example, Aguayo et al. (2023), who tuned the

parameters of a simpler temperature index model in OGGM for all of the glaciers in the Patagonian Andes. A more efficient calibration routine, or the availability of sufficient computational resources would be required to calibrate the JULES parameters in this way. However, this approach would need careful consideration of how this model could reasonably be "validated" outside of its calibration data. One could also argue that the inherent uncertainties in geodetic mass balance data, particularly for the smallest glaciers (Dussaillant et al., 2019), could lead to significant model biases. In the same vein, it is fair to assume that at least some of the apparent model inconsistencies are themselves, manifestations of the uncertainty of the geodetic validation data. While this may be true, the assessment of the model fit to the 30 calibration glaciers, which were chosen for their relatively large size, showed similar levels of inconsistency which suggests the overall impact on interpretations of model deficiencies is minimal. In fact, the use of the geodetic data for calibration, we would argue, is more justifiable than using sparse glaciological data for this tropical setting. Indeed, when Shannon et al. (2019) calibrated JULES using a similar approach to that used here, but with elevation-band specific mass balance observations from the World Glacier Monitoring Service (Wgms, 2023), their model showed an unrealistically-negative bias for low-latitude regions.

Other improvements to the calibration strategy that could provide gains in model performance include a more suitable grouping of glaciers, that better-captures the differences in glacier properties (and model parameters), instead of the location-based regionalisation implemented in this study. What the basis would be for this grouping, however, is not clear from this study given that there was no clear coherence between simulated mass balance errors and glacier characteristics. One should also consider that the ice flow parameters in OGGM were not considered in the calibration procedure. This has obvious implications for the simulated ice dynamical response to climate forcing and for the initial ice thickness inversion step, both of which have the potential to introduce errors into the forward projections. The recent work of Aguayo et al. (2023) has shown how the OGGM creep parameter can be included in the calibration of OGGM. They propose a method whereby the this parameter is selected so that the ice thickness inversion step, used to identify the initial ice volume, matches that of an independent set of glacier volume estimates similar to those of Millan et al. (2022). Within their calibration routine, the model is further validated against RGI glacier outlines for the year 2000 by implementing a "dynamic spinup" of the model from 1980. Only when the model can capture the RGI glacier area, the glacier volume estimates and geodetic mass balance data is the calibration deemed a success. Implementing this type of calibration approach with JULES-OGGM is feasible, although comes with its own computational constraints that would not make this straightforward. Even so, it could be that additional gains in model performance are found by considering the ice dynamical parameters in the model calibration.

While modifications to the calibration strategy may be beneficial to model performance, the expected benefits can only be postulated here. This study suggests that meaningful improvements in model performance could be attained through targeting improvements to the prognostic snow albedo routine in JULES. Specifically, it appears that the model cannot accurately replicate the feedbacks between the driving meteorology, surface energy balance, ablation processes and snow darkening. Because of this, the calibration problem was dominated by the sensitivity to the parameter that controls the weighting of albedo between visible and near-infrared wavelengths (wght_alb) in JULES: a parameter that is important in controlling the rate at which snow albedo drops as the snow ages and grain size increases. While this parameter can be tuned to capture long-term

average estimates of glacier-wide mass balance, comparisons of simulated and observed surface albedo at the Quisoquipina meteorological station revealed an apparent case of "correct answer for the wrong reason": a case that is likely to manifest itself in inconsistent performance outside of the spatiotemporal bounds of the calibration dataset. Only by forcing this aspect of the model with observed net radiation variables, were we able to more-accurately capture day-to-day and seasonal variability

in surface albedo and the observed ablation rates during the main ablation season. The prognostic snow albedo routine in JULES is based on the Wiscombe et al. (1980) spectral snow model which simulates changes to snow albedo, both in the visible and infrared wavelengths as a function of effective snow grain size. The models of snow grain size development and spectral albedo are, in essence, empirical and validated against a small number of observation studies in the Antarctic and Canadian Archipelago which raises the question of their validity in other settings. They also highlight two limitations of the

model in the inability to account for: i) snow impurities; and ii) scattering properties of non-spherical snow grain shapes. Ice cores drilled from the Quelccaya ice cap on the eastern edge of the VUB show seasonal deposition of mineral dust during the dry season when westerly winds facilitate entrainment and transport from the dry Altiplano (Thompson et al., 2013). (Reis et al., 2022) found that between the years 2002 and 2018, the average dry season dust concentration was $6.8 \pm 1.8$ ppm but could be as high as 23.1 ppm. A limited number of observation studies suggest that black carbon concentrations of less than 1 ppm

can reduce snow albedo by several percent, but that the effect of mineral dust is approximately two orders of magnitude smaller (Willeit et al., 2018). However, recent numerical modelling studies suggest that the impact of mineral dust on snow albedo could be much more significant and is strongly influenced by snow grain size and non-sphericity. He et al. (2019) explicitly simulated the deposition and redistribution of dust through the snowpack as well as the scattering properties of non-spherical snow grain shapes in the stochastic aerosol-snow albedo model (Liou et al., 2014) and showed that, in the visible wavelength

spectrum, snow albedo can be reduced by almost 20% when dust concentration is 100 ppm and snow grain radius exceeds 1000 µm although the magnitude of reduction is dampened for non-spherical snow grains. Hao et al. (2023) demonstrated that including these processes in the E3SM land surface model, allowed them to more closely-match observed albedo dynamics on the Tibetan Plateau. However, they highlight that the combined effects of non-sphericity and snow impurities on albedo are complex, nonlinear, and may be negative or positive depending on the approach used to parameterise them. The composition

of other impurities e.g., biological communities which could also serve to reduce snow and ice albedo (Hotaling et al., 2021), have to our knowledge not been observed or quantified in the study region. So, while gains could be made through implementing improvements to the prognostic snow albedo routine in JULES, such improvements would need to be underpinned with by appropriate observation data to appropriately parameterise them. Meeting this challenge will undoubtedly require improvements to our process models, but likely also requires us to better-constrain mountain snowfall timing and

frequency which has a significant influence on albedo dynamics (Johnson et al., 2020).

## 4.2 The "ice-free" limitation

In the methodology, we noted that the JULES-OGGM sequential coupling does not explicitly account for differences between surface energy balance fluxes on ice-free and ice-covered surfaces which could be important when OGGM determines if a

glacier will advance into an ice-free node on a given time step. For ice-free nodes, OGGM takes the sum of the surface mass
balance and ice flowing into that node from upstream. If this sum is positive, the glacier "grows" into this node. With the
presented JULES-OGGM setup, the surface mass balance for the ice-free node is calculated as though it were ice-covered.
This means that there could be some situations where JULES-OGGM simulates the advance of a glacier at a given timestep
when an identical model that accounts for the ice-free energy balance does not (and vice versa). As a thought experiment we
could hypothesise a JULES-OGGM model that has an equivalent set of ice-free JULES gridbox nodes for each glacier spanning
the full elevation range of the glacier. We could then force OGGM with surface mass balance simulations that are tailored to
the ice coverage conditions at a given point in time. The overall impact that using such a model would have on the simulation
is difficult to postulate given that it will depend a lot on how the exposed ground is parameterised in JULES, but we would
hypothesise that sensitivity to this model limitation would be most pronounced for i) advancing glaciers where the surface
mass balance will impact the ice thickness at the margin during transition from ice-free to ice-covered; and ii) glaciers where
simulated mass balance at the tongue is close to zero and therefore, small changes in the simulated mass balance on ice free
nodes could decide whether or not a glacier grows at a particular timestep. For glaciers undergoing sustained retreat where the
surface mass balance at the glacier margin is consistently negative, the choice of using JULES-OGGM with or without our
hypothetical improvement would have no discernible effect on the simulation. For this reason and the fact that the mass balance
of the glaciers in the VUB over the simulation period is negative (particularly at the glacier tongue), we would argue that
discrepancies introduced by the ice-free limitation are likely to be small in this study.

## 4.3 Twenty-first century simulations

Considerable variability in glacier response to climate exists across the VUB. We identified four broad mass evolution clusters
ranging from glaciers that lose most (if not all) of their mass before 2040, to those that showed an initial mass gain before
subsequently retreating through the latter-half of the 21$^{st}$ century. While the exact trajectory of the simulations needs to be
considered within the limitations of the model, the results suggest that the dominant control on this variability is the glacier
geometry. Specifically, the smallest glaciers at low and high elevation ranges represented the most and least rapidly retreating
glaciers respectively. Fyffe et al. (2021) also found that elevation plays an important role in controlling the non-linear response
of tropical glaciers to changes in air temperature. At higher elevations, they suggest that decadal changes in mass balance are
more-strongly-controlled by the transition of sublimation fluxes to melt. While this may be true at very localised high-elevation
ice, we did not find any evidence of significant sublimation fluxes at the river basin scale. In fact, for all identified clusters,
sublimation showed to have a negligible contribution to annual glacier mass changes throughout the simulation period. This
does raise the question on the gains achieved through applying energy balance models for basin-scale analyses like these and
whether we should sacrifice the level of fit that can be achieved with a simpler temperature-based ablation model just for the
sake of providing a more physically-coherent model of the system. We suggest that this depends on the purpose of the
application and the region of interest. The apparent insignificance of sublimation in the VUB should not be expected
everywhere. Furthermore, our simulations still showed deviations between long term temperature and mass balance

trajectories. Specifically, the shortwave radiation showed to be the dominant energy source across the glaciers, which is controlled by the surface albedo which was not necessarily coherent with temperature. Given the importance of shortwave radiation, a key limitation of the projections is that this variable was effectively assumed static for the future simulation period, and therefore, the projections do not capture the impact of changes in the radiation balance in the future. To explore this limitation in more detail, we extracted the incident long and shortwave radiation data from each CMIP5 ensemble member at the point closest to the middle of the domain (at -13.69ºN, -72.02ºE). They showed that the annual average incident longwave radiation is projected to increase by an average of 6.63% from the 1980-2018 average to the 2061-2099 average under RCP8.5, with all models showing a statistically significant increase. They also showed an increase in average incident shortwave radiation by 3.20 % over the same time period with 22 models showing a statistically significant increase, 4 showing a statistically significant decrease and 4 showing no significant change. While these changes would only result in a relatively small change to the radiation balance, they are not accounted for in our projections.

To provide some context for the simulated ice area and mass simulations, we obtained simulations for the VUB from the GlacierMIP experiments (Marzeion et al., 2020) for the GloGEM (Huss et al., 2015) and MAR2012 (Marzeion et al., 2012) models. These models are much simpler than JULES-OGGM, using temperature-based energy balance models and empirical ice evolution routines to account for mass redistribution. When driven with the same CMIP5 GCMs, the overall trajectory of total glacier area and mass is broadly similar across the three models (Figure C 1) showing a consistent decrease over time. However, JULES-OGGM shows to be more conservative (slower retreat), particularly with respect to the change in glacier area. By 2100 under RCP4.5, JULES-OGGM estimates that 17% of the ice mass will remain by 2100, while GloGEM and MAR2012 estimate this to be 2%. For RCP8.5, JULES-OGGM estimates that 6% of the ice mass will remain, while GloGEM and MAR2012 predict that less than 1% will remain. These differences likely stem from a range of sources in addition to differences in the glacier models themselves. Arguably of major importance, is the fact that JULES-OGGM was shown to overestimate surface albedo, at least at the one observation point in the basin. If this behaviour is indicative of basin-wide simulations, the apparent conservativeness of the simulated retreat likely stems from this model deficiency. Additionally, as has been discussed, we did not use the raw CMIP 5 climate projections used to force the GloGEM and MAR2012 projections. Instead, we used a combination of statistically-downscaled and resampled driving climate data which will have inherently different biases and trends over the twenty-first century simulation period. Aguayo et al. (2023) compared the impact of perturbing various aspects of the model setup for OGGM on glacier runoff simulations for the Patagonian Andes and showed that the uncertainty in simulated glacier runoff mainly stemmed from the choice of historical climate data used to calibrate the model for 78% ± 21% of the catchment area. In contrast, "future" sources of uncertainty (choice of climate model, emission scenario and bias-correction method) were only the main sources of uncertainty for 18% ± 21% of the catchment area. Similarly, Li et al. (2022) showed that for central Asia, the projected mass loss difference between two identical glacier dynamics models with different initial glacier inventories was higher than that of adjacent emission scenario forcing data. Aguayo et al. (2023) also showed the significance of other aspects model setup such as the initial ice thickness and glacier outline maps. Indeed, our study used the more refined glacier outline maps of Drenkhan et al. (2018) than those in the

GlacierMIP and there are clearly differences in our initial glacier areas which we estimate as significantly smaller (186.5 km$^2$) compared to those from the GlacierMIP models which are based on the less-accurate RGI 6.0 glacier outlines (250.1-264.3 km$^2$). For the initial ice volume, it is important to note that our simulated ice mass for the year 2000 lies somewhere between that of GloGEM and MAR2012 (Figure C 1c and d) which suggests that this is not a principal driver of the differences in our

simulations to these models. Certainly, when we compared our ice volume simulations for 2018 against the estimates of Millan et al. (2022) the level of agreement was encouraging, but there were some discrepancies, especially for the smallest glaciers where the JULES-OGGM simulations generally underestimate glacier volume. Even so, there are significant uncertainties in the Millan et al. (2022) ice volume estimates, given the temporal mismatch in input data. They also note that errors in ice thickness (and thus volume), for thicknesses below 100 m, which accounts for 95% of the glacierised area in the VUB, are

assumed to > 50%.

Perhaps most-significantly, we tuned JULES-OGGM to geodetic data, while the GlacierMIP models were tuned to World Glacier Monitoring Service (WGMS) data. Indeed, this could explain why, even though the model evaluation showed JULES-OGGM to overestimate mass loss by a factor of two, it still shows to be more conservative than the GlacierMIP models. A crucial component in understanding how to improve physically-based glacier evolution models like JULES-OGGM will be

developing a fuller understanding of where the sensitivities (and potential sources of error) in the model setup lie. Implementing a framework akin to Aguayo et al. (2023) using JULES-OGGM should be a priority for developing this understanding. One should not expect the relative importance of different sources of uncertainty to be the same. Including more complex models like JULES-OGGM in large-scale model comparison studies like GlacierMIP where input and forcing data are standardised would also help to deduce sources for model discrepancies. The GlacierMIP results have already shown that the choice of

glacier model can be the dominant source of uncertainty in projections, particularly for the low-latitudes, highlighting the need to develop and test different modelling approaches (Marzeion et al., 2020). Indeed, both JULES and OGGM have been applied at the global scale, and while this study has identified potential aspects that should be prioritised for application to tropical glaciers, it remains to be seen how this model performs in other glacierised basins around the world.

## 5 Conclusions

We show that the model parameters in JULES-OGGM can be tuned to capture observations of long-term average glacier mass changes, as determined from geodetic glacier mass balance data, but that inconsistencies in model performance can be attributed, at least in part, to limitations of the JULES prognostic snow model that cannot accurately replicate observed fluctuations in surface albedo which has important implications for the radiation balance of snow and ice. This study suggests that a key challenge in applying "physical" glacier evolution models at the scale of whole river basins in tropical regions lies

in our limited understanding and ability to represent surface albedo dynamics. We suggest that this should be a priority development area for future applications of JULES-OGGM. Two key limitations of the current snow albedo routine are the lack of: i) representation of deposition and redistribution of snow impurities and ii) parameterisation of non-spherical snow

particles. Furthermore, we have not examined the role of the ice flow component in the identified model deficiencies and so additional improvements could be made by, for example, including the ice flow parameters with the model calibration step.

The results from this study also indicate that, contrary to point-scale energy balance studies, at the basin scale, sublimation will likely play a very minor role in the evolution of glaciers in VUB over the twenty-first century and will not be a significant source of non-linearities in the glacier response to climate warming. These results are not necessarily indicative of all glacierised basins in the tropics, but do imply that sublimation processes may not be as important for long term glacier evolution as some studies suggest. Indeed, we believe there is much to be learned from applying a physically-based, globally-capable

model like JULES-OGGM to other basins inside and outside of the tropics and the availability of global geodetic datasets provides an opportunity to interrogate and validate the model for any almost any glacier in the world.

## Appendix

### Appendix A: Lapse rates

The plots below show the annual lapse rate cycle for all years of historical data (1980-2018) for each calibration region of
glaciers in Figure 1. The lapse rates were first estimated for each glacier individually based on the climate simulations from the four bounding WRF nodes of the glacier centroid. The median of these was then taken for each calibration region.

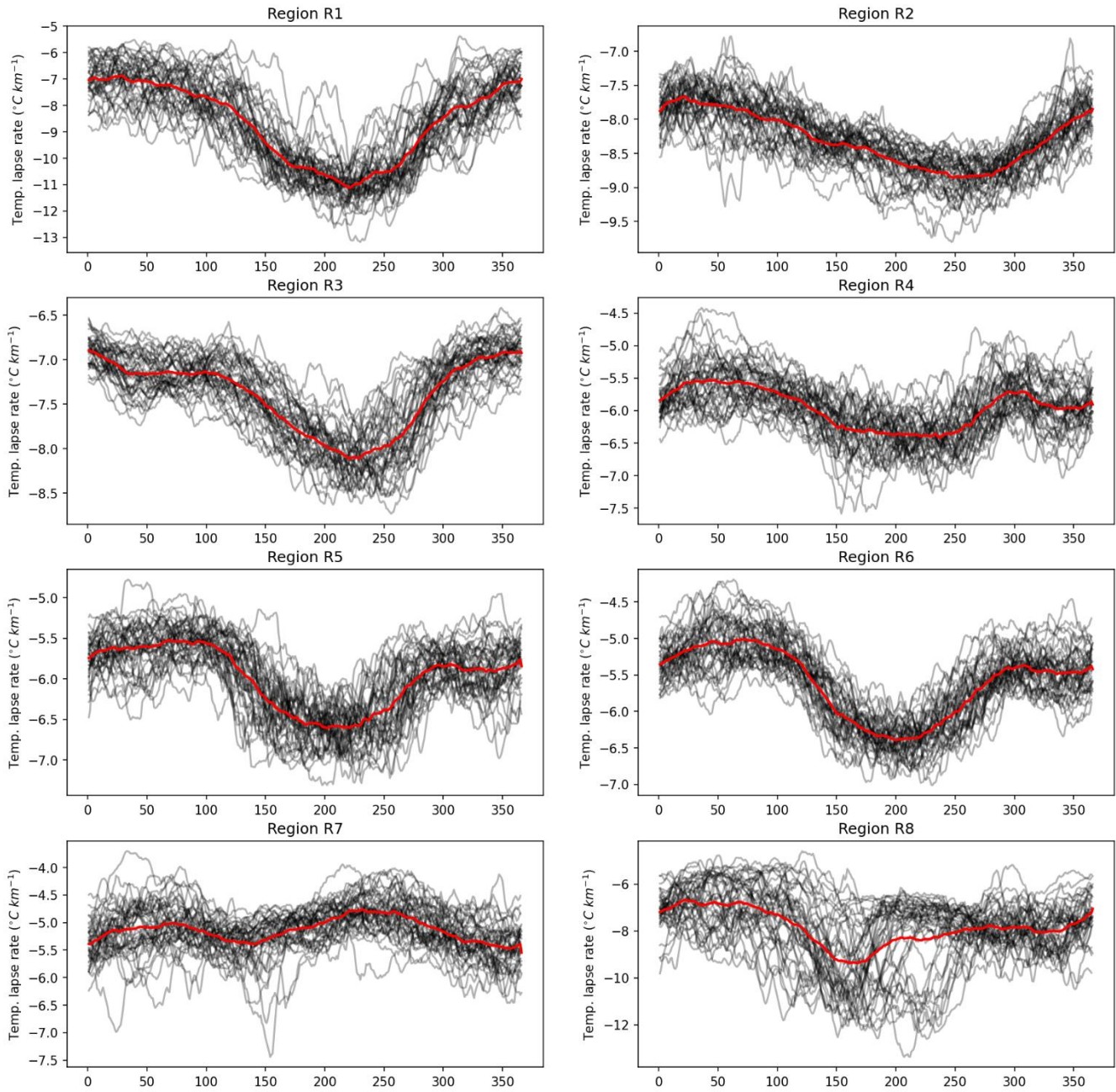

**A 1: Annual temperature lapse rate cycles for all historical years (1980-2018, black lines) overlain with the median annual cycle (red line).**

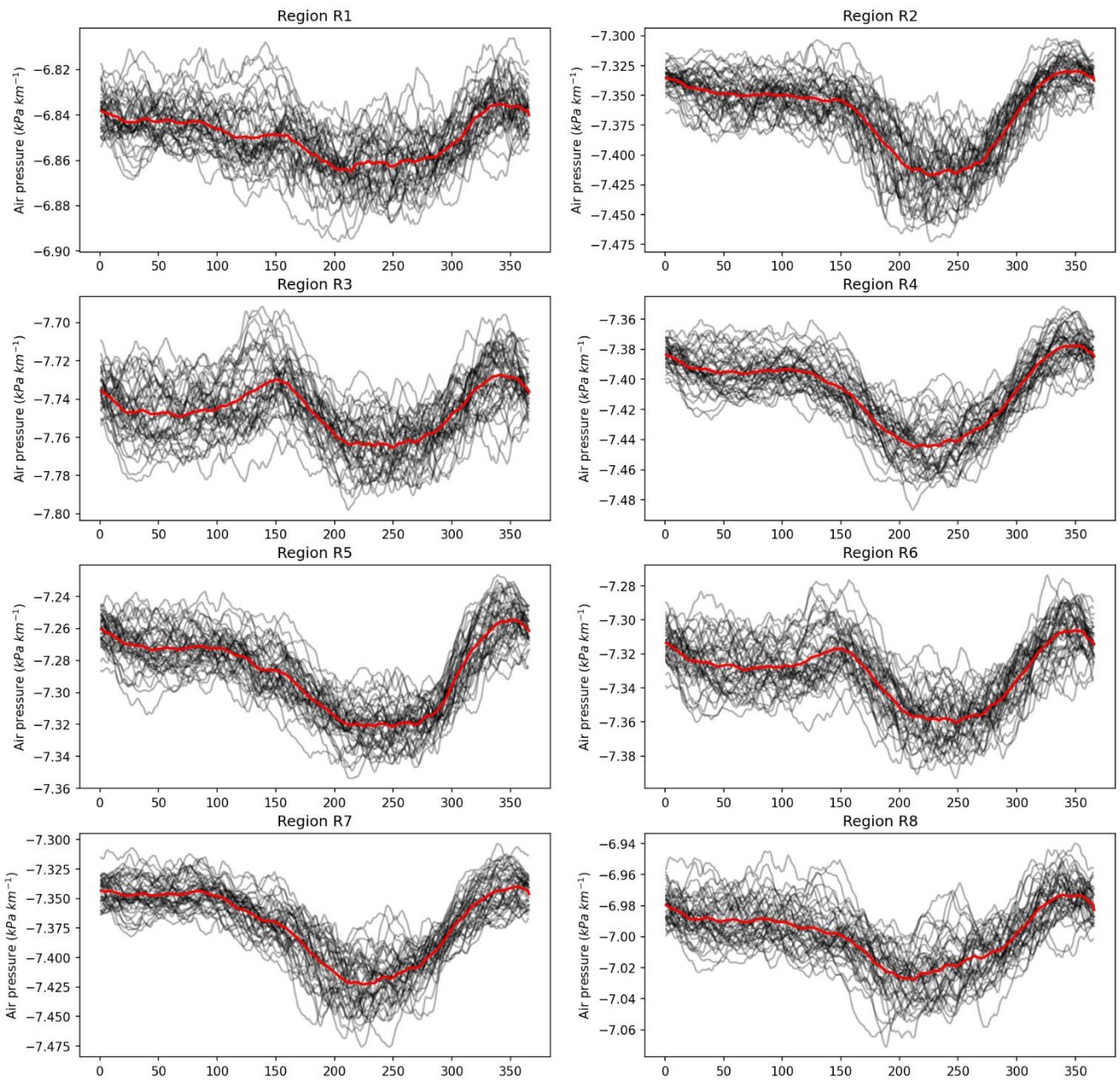

A 2: Annual surface pressure lapse rate cycles for all historical years (1980-2018, black lines) overlain with the median annual cycle (red line).

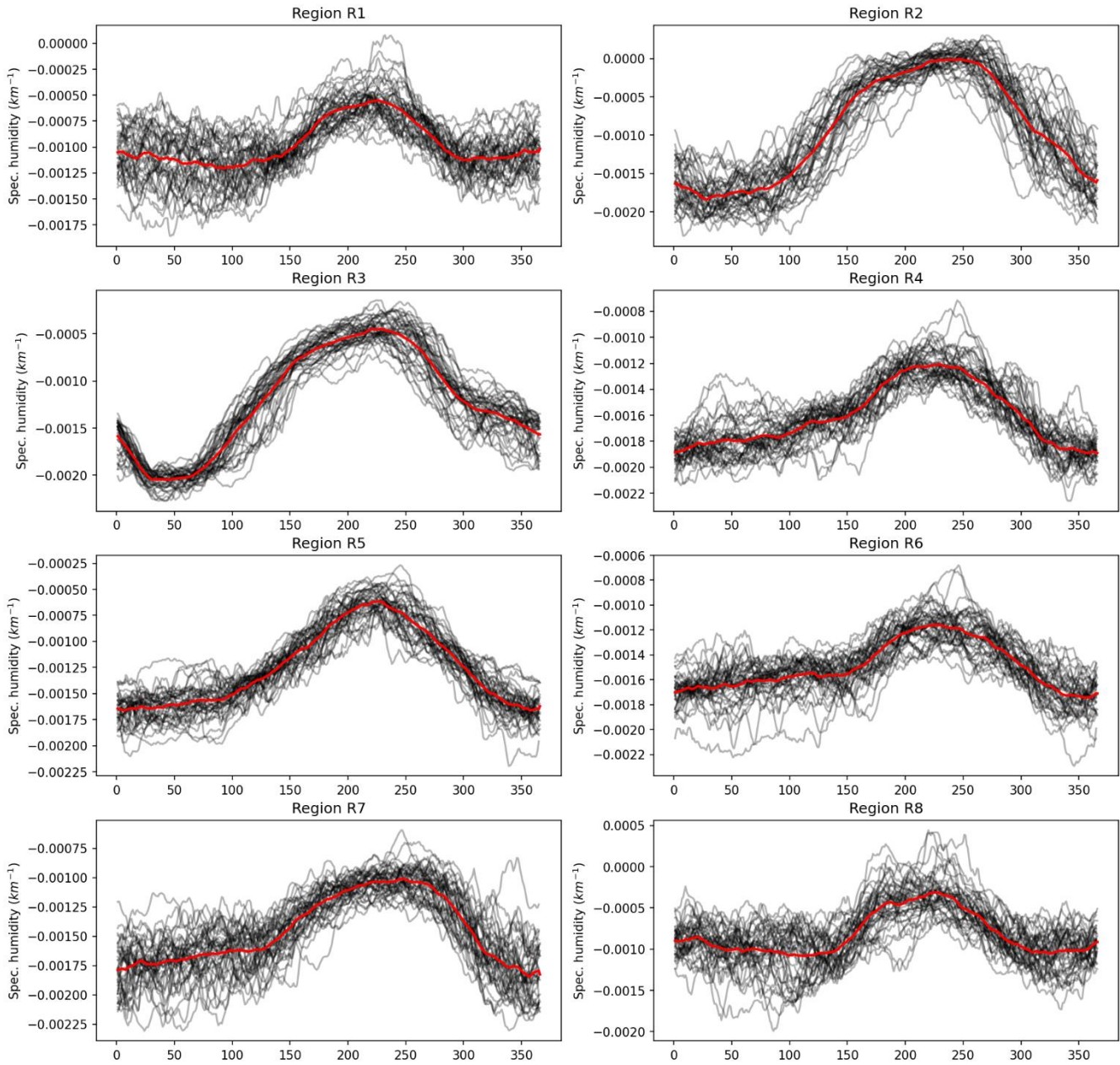


**A 3: Annual specific humidity lapse rate cycles for all historical years (1980-2018, black lines) overlain with the median annual cycle (red line).**

 **Appendix B: Assessment of relationship between glacier characteristics and simulated mass balance errors**

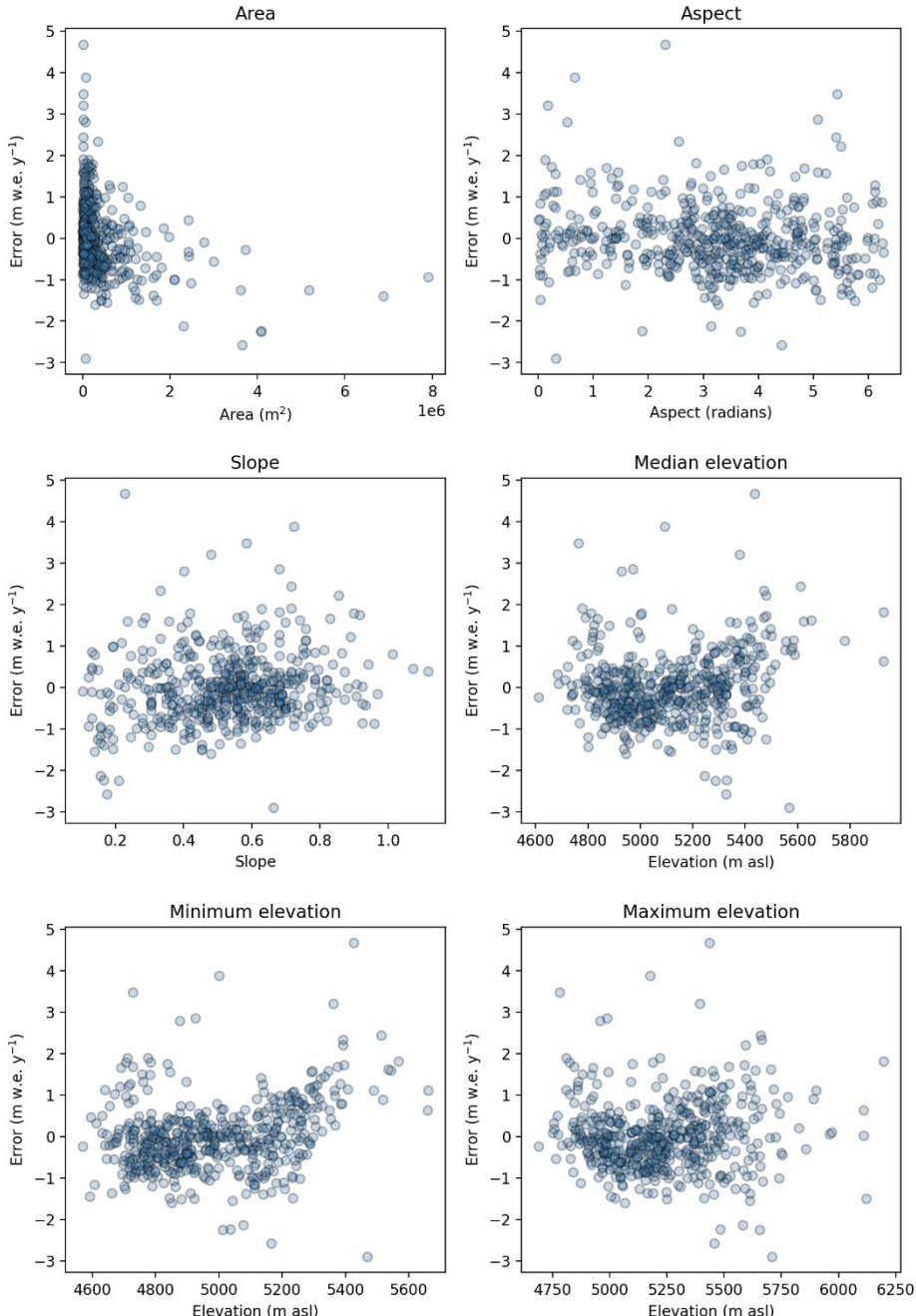

**B 1: Comparison of simulated mass balance errors to six glacier characteristics.**

**Appendix C: Comparison of JULES-OGGM simulations to subset of GlacierMIP simulations**

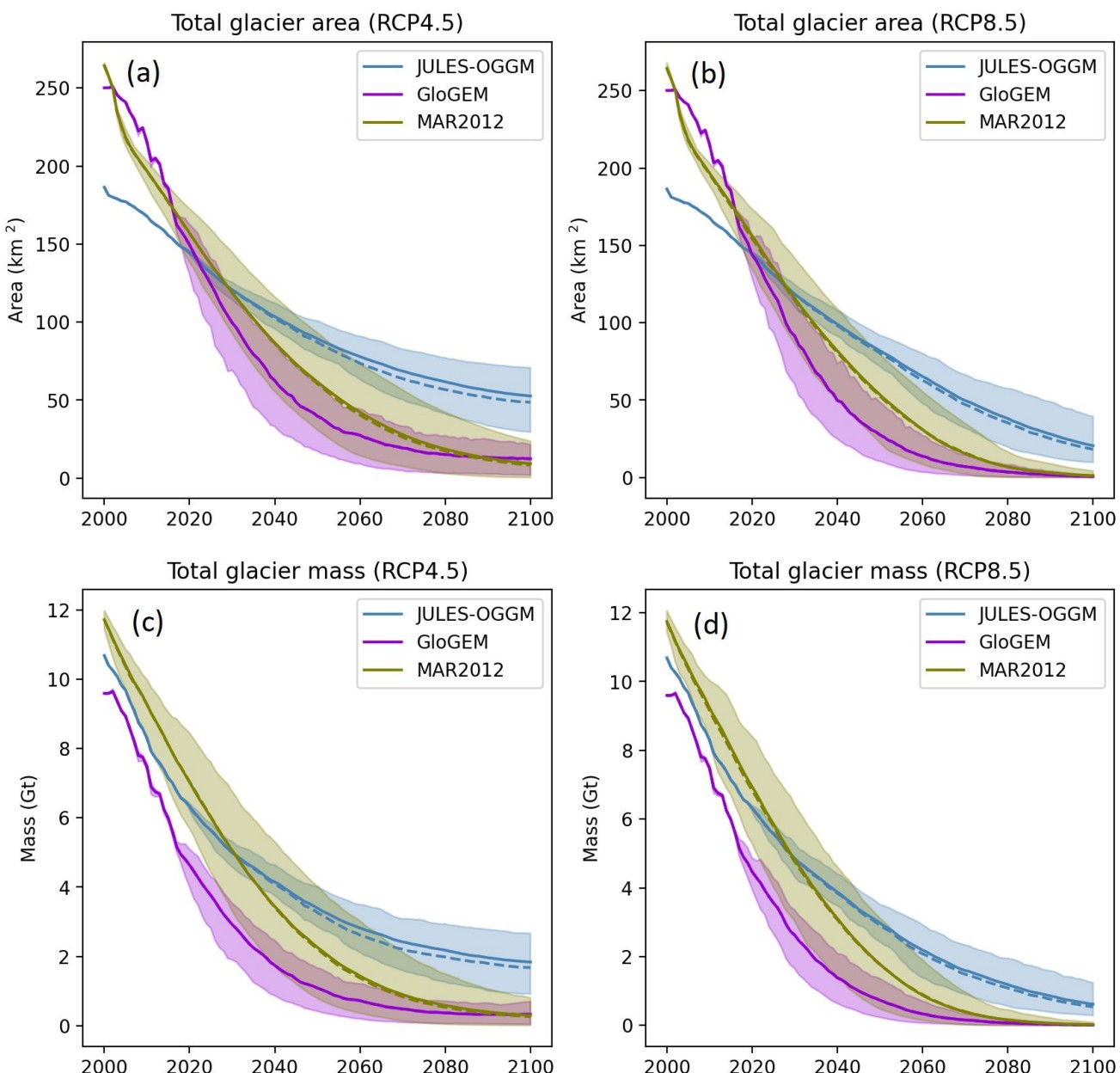

C 1: Simulated annual total glacier area (a,b) and mass (c,d) for the VUB between 2000 and 2100 based on RCP4.5 and RCP8.5 scenarios. Simulations include JULES-OGGM and those from the GloGEM and MAR2012 models from the GlacierMIP experiments (Marzeion et al., 2020). Solid lines represent the ensemble mean simulations using all available GCMs (not coherent across glacier models. Dashed lines represent simulations driven with the subset GCMs that are coherent across the glacier models. Shaded areas represent the 90% confidence bounds.


## Appendix D: Selection of BNU-ESM simulation

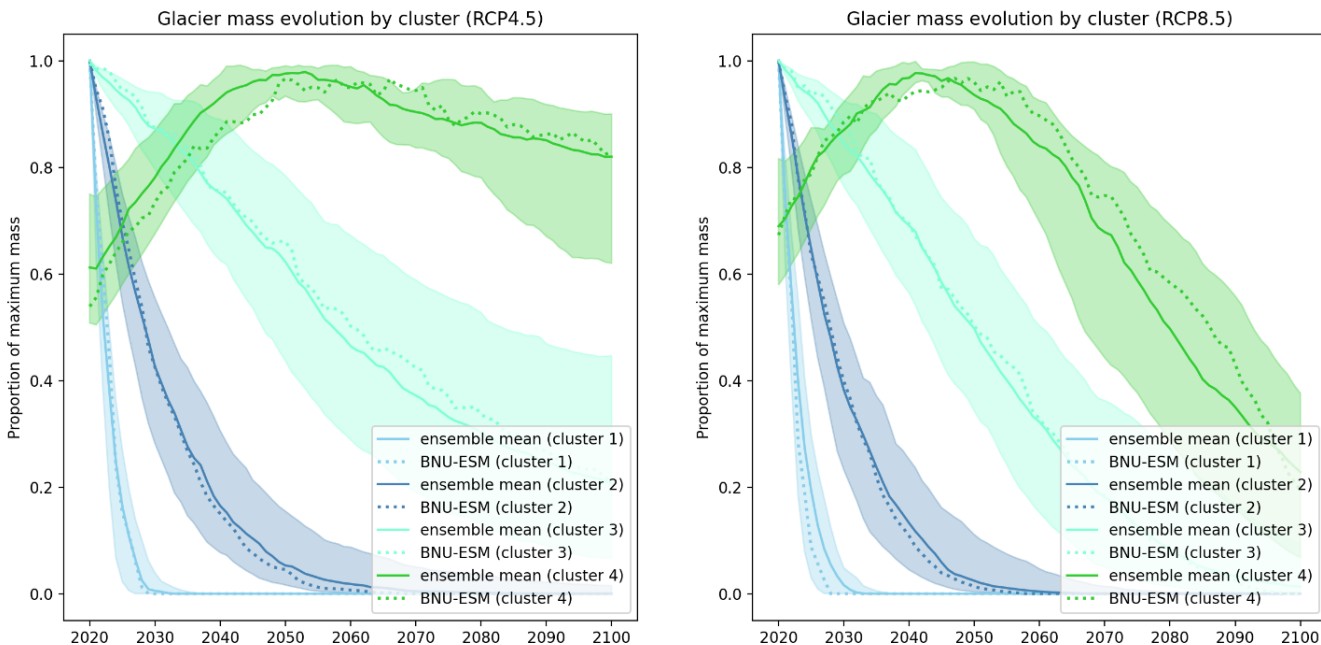


**D 1: Annual glacier mass evolution of four identified clusters expressed as the proportion of the maximum simulated mass between 2020 and 2100 for RCP4.5 (a) and RCP8.5 (b) overlain with the BNU-ESM simulation which was selected to most-closely match the ensemble mean. Annual glacier mass evolution of four identified clusters expressed as the proportion of the maximum simulated mass between 2020 and 2100 for RCP4.5 (a) and RCP8.5 (b) taken from the ensemble mean. Each cluster is represented by the**
**median (solid line) and interquartile range (shaded area).**

## Appendix E: Mass and energy balance simulations under RCP4.5

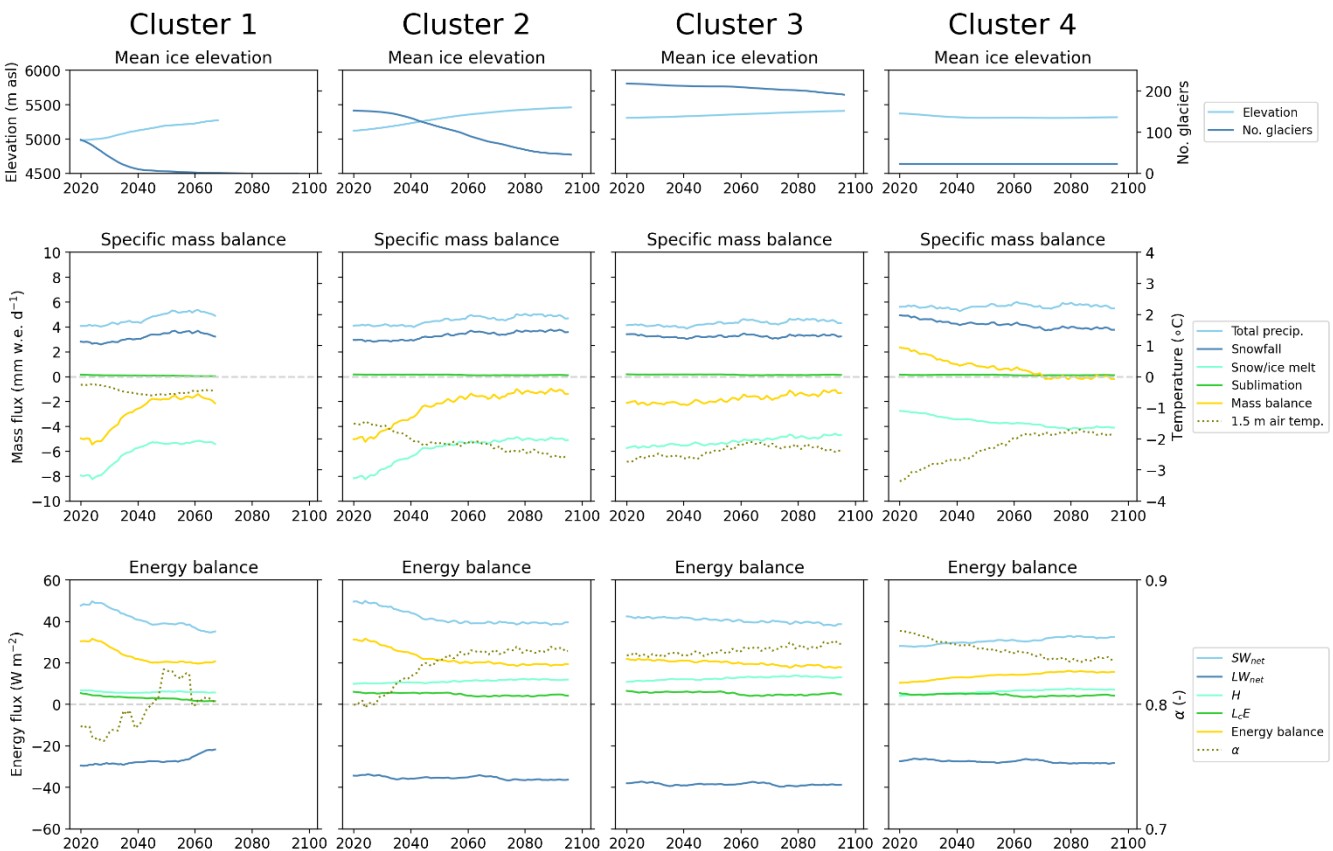

**E 1: Simulated mean glacier-wide state-variables from JULES-OGGM between 2020 and 2100 for each cluster (columns) and based on the BNU-ESM simulation under RCP4.5. The variables include ice elevation (top row), specific mass balance (middle row) and energy balance (bottom row). All variables are calculated over a 10-year moving average window and are taken as the mean across all glaciers within a cluster, weighted by the glacier area.**

## Code availability

The source code for JULES v6.0 can be downloaded by accessing the Met Office Science Repository Service (MOSRS) (requires registration): https://code.metoffice.gov.uk/. All code used to run the JULES experiments in this study are also available on MOSRS and are stored within two versioned JULES Rose suites: The Monte Carlo experiments are stored in the u-ce887 JULES Rose suite and the future climate experiments are stored in the u-ck523 JULES Rose suite. The model runs at the Quisoquipina and Suyuparina glaciers are stored in the u-cw985 JULES Rose suite. The source code for OGGM v1.4 used in this study can be accessed via Maussion et al. (2021). The safepython python library version 0.2.0 used in this study (Pianosi et al., 2015) and Pysolar v0.11 are both available to use for free under the GNU general public license.

## Data availability

Data are available upon request to the corresponding author.

## Author contribution

JDM led the code development, glacier modelling, analysis and writing of the manuscript. EP led the climate modelling, bias-correction and downscaling for input to JULES-OGGM. All other authors contributed to the development of ideas and writing of the manuscript.

## Competing interests

NEB is a member of the editorial board of The Cryosphere. All other authors declare that they have no conflict of interest.

## Acknowledgements

This work was jointly funded by the NERC-UKRI WateR security And climate cHange adaptation in PerUvian glacier-fed river basins (RAHU) (Grant number: NE/S013210/1), the Hydro-JULES research programme supported by NERC National Capability funding (Grant number: NE/S017380/1) and the TerraFIRMA project supported by NERC National Capability funding (Grant number: NE/W004895/1). JDM publishes with the permission of the executive director of the British Geological Survey.

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
