# Peer review of "Physically-based modelling of glacier evolution under climate change in the tropical Andes"

_EGUsphere, 2024_

## Referee Comment (RC2)

Review of the article entitled:

**"Physically-based modelling of glacier evolution under climate change in the tropical Andes"**

In this study, the authors present a framework to *sequentially couple* the ice dynamical part of the Open Global Glacier Model (OGGM) with the full energy balance model for snow and ice from the Joint-UK Land Environment Simulator (JULES). The authors apply this sequential (offline) coupling to 500 glaciers in the tropical Andes and make projections of glacier mass loss until 2100 for different RCP scenarios. They conclude that under RCP4.5 17% of the ice mass will still remain by 2100 which is more than what other glacier modelling studies have predicted e.g., compared to 2% as predicted by GloGEM (Huss and Hock 2015) and Marzeion et al. 2012.

The authors present a very unique and clever workflow which consists of running both models separately; JULES is in charge of computing the annual specific mass balance over discrete points in the study domain. Thus, this model comes up with different relationships of surface mass balance as a function of height per year $SMB(z_i)_t$ for each glacier. OGGM then extracts the annual specific mass balance at a particular location, elevation and time. The ice dynamical flowline module of OGGM then inverts for the ice thickness via mass conservation and the Shallow Ice approximation (SIA). For the glacier evolution OGGM solves the continuity equation with the updated SMB distribution given by JULES.

The authors only calibrate parameters within JULES full energy balance model. Ice dynamical parameters in OGGM are not calibrated. JULES SMB calibration consists of several steps where JULES parameters are modified to achieve the best fit to geodetic mass balance data for a benchmark of 30 glaciers; then parameters are extrapolated to other glaciers within small subregions.

Overall, the manuscript is well written and the methods and discussion section has a clear narrative and description of model experiments. It is clear that the authors have put a lot of effort in model calibration and evaluation of SMB. However, some parts of manuscript lack order and could be re-arranged slightly to enhance the impact of the results and discussion section. There are certain aspects of the methods that require clarification and the discussion section neglects to highlight certain limitations of not updating changes in glacier geometry in JULES simulations. Also, the authors do not asses the implications of not calibrating ice dynamical parameters in their simulations.

I will definitely recommend the publication of the manuscript after the authors clarify some of my questions below and make some changes to the manuscript. I also recommend below how the authors could evaluate their framework limitations regarding ice dynamics.

Major comments:

- The JULES model is unaware of changes in the glacier hypsometry through time i.e., JULES is not aware of glacier retreat simulated by OGGM. Glacier retreat might affect the $SMB(z_i)_t$ relationships that JULES outputs; if at a specific height $z_i$ and location that OGGM node becomes ice-free in a given year. Authors argue that changes in glacier hypsometry through time are accounted by lowering the ice surface and feeding into OGGM an SMB at a lower elevation. How this is decided between timespans and how authors deal with the transition between ice and ice-free areas is not clear. For example, given two points along the flowline, the $SMB(z_i)_t$ relationship might not hold if the second point (at a lower elevation) becomes an ice-free area. Given that ice free physics are qualitatively different than ice covered areas the interpolation might no longer hold. Limitations regarding this issue are not explained by the authors and this should be clarified in the explanation of step 2 in section 2.4.3. and the discussion part.

- The simulated glacier geometry that results from the interaction between mass balance and ice flow processes should be compared to ice thickness or volume observations. This is important as the initial glacier state can have a large impact on the simulated glacier evolution (Zekollari, et al. 2022). Authors do not validate the initial ice thickness distribution obtained with their JULES-OGGM framework before doing future simulations. They only evaluate the model by comparing simulated SMB to geodetic and in situ SMB observations.
There might not be in situ glacier thickness measurements for these glaciers, however, the authors could check how their simulated initial ice volumes or ice thickness distributions compare to those from Millan et al (2022) at least for glacier wide volume estimates. Millan et al (2022) is a satellite derived ice thickness product and uses, like OGGM, mass conservation and the SIA, these are not in situ ice thickness observations but if compared to model initial model simulations could provide an idea of the calibration error for the ice dynamic part of the workflow. In other words, errors derived by not calibrating ice dynamical parameters in OGGM or by not updating glacier geometry changes in JULES. By looking at Millan et al (2022) Figure 3b seems to be data available for C Vilcanota glaciers. I encourage the authors to make such a comparison as this could strength the findings of the manuscript.

Minor changes:

**Abstract:**

L26-27: "We conclude that this inhibits the robustness of extrapolating the JULES parameters across multiple glaciers". I will remove this from the abstract because this is true for any glacier model, parameters describing specific aspects of each glacier can't be extrapolated to other glaciers. See calibrations sections from Marzeion et. al. (2012) and Zekollari, et al. (2022).

Authors should highlight the societal importance of their work in the Abstract and how important this region is for water availability.

**Introduction**

L44. Replace The is … with "That is especially true in…"

L49. "Sublimation" add citation and definition as this is the first time the authors mention the concept.

L49-50: "sublimation can account for the majority of energy consumed for ablation" change to "can reduce the energy available for melting" (clearer and in line with Winker et al. 2009).

Add at the end of the introduction the key takeaways from coupling these two models.

**Methods**

My main suggestion here is to change the outline in the following way:

2.1 Study Area. This section should be moved and integrated into the Introduction, here authors could make the point that a lot of people depend on these water resources (see Millan, et al. 2022 figure 3b) and given the little knowledge surrounding this area, the authors work is highly important.

Sections 2.2 – 2.3.
Should be a new section called: **2. Input data and pre-processing**. There authors should explain all the data input used in the model and the pre-processing stages needed to ingest the data into JULES-OGGM workflow.
E.g., The authors used their own Glacier inventory. The processing of these outlines and why they choose those instead of the Randolph Glacier Inventory could be specified in a subsection e.g., 2.1. Glacier outlines.

With this format authors could also expand into the analysis done to the climate input data.

2.3 Glacier Mass Balance data: Specify the advantages of Dussaillant et al (2019) vs Hugonnet, R and others (2021)? Most glacier modelling studies use Hugonnet, R and others (2021) to calibrate parameters in the glacier mass balance. E.g.  Rounce et al. 2023.

2.4 JULES – OGGM glacier modelling workflow
Here I think authors could make a simple change to make the outline of the methods more organised:

3. Methods: glacier modelling workflow

3.1 JULES

3.2 OGGM

3.3 Sequential (offline) coupling of JULES-OGGM

This change will make it clear that the coupling is sequential and that both models are not fully coupled but one feeds input to the other.

L179. Replace "they" with Shanon et. al (2017).

L194. Replace "the flowline model" with "ice dynamical flowline model"

L194. Replace ": in effect," with "– i.e.," …

L199. Point the reader to the Figure 3a.

L202-203. Climate data pre-processing details could be moved to the input data and pre-processing section.

L252. Add citation for the Randolph Glacier Inventory (RGI). Move that part of the workflow to the Input data and pre-processing section (e.g., add a Glacier outlines section).

Here the authors should mention the implications of not using the RGI, which limits the comparison of JULES-OGGM results with previous model estimates that use the RGI. e.g., Li, F., et al. (2023) found that in High Mountain Asia projected mass loss differences between inventories are higher than between adjacent emission scenarios, illustrating the vital importance of high-quality inventories.

This might not be as relevant for the Andes but having an idea of how different they are in total area coverage per glaciers should justify why authors do not use the RGI when comparing JULES-OGGM results to GloGEM and Marzeion et al. 2012 in Appendix C1.

Authors should add a calculation of how different the Area coverage per glacier is between the RGI and their inventory of choice.

L257 When describing OGGM equilibrium assumption. Authors should specify the OGGM version used in their study. (e.g., v.1.4) as that assumption is not required by OGGM in the latest version (see model updates and documentation).

L268 "while the default parameters for the ice flow component of OGGM were used." Add the implications of this in the discussion and limitations, point to that section.

L271 "except for the temperature lapse rate" Add (See Appendix A).

L280. Add citation to the Monte Carlo framework used and add citations for other studies which have used the same strategy.

L294. "Used across environmental modelling applications". Add citations.

L298. Add safepython library citation, version used.

**Results**

L307-308. Fig 4a shows a perfect correlation, I wonder if multiple parameter combinations could achieve the same thing?

**Model Evaluation**

L347. The authors could enhance the discussion and provide an explanation from where the errors come from. This could be down to several limitations in their approach:

i) The JULES full energy balance model is not aware of changes in glaciers hypsometry through time. Thus, the SMB(z) relationships do not incorporate well ice dynamical feedback from OGGM. In other words, lowering the ice surface might not be a realistic representation of glacier retreat.

ii) the ice dynamical parameters in OGGM are not calibrated per glacier. Is it known that ice thickness (and ice volume) decreases as a function of the A factor; thus, perhaps the default parameter in OGGM might not be the right value for these glaciers? See Maussion et al. (2019).

iii) A comparison between the ice thickness obtained with JULES-OGGM vs Glacier thickness observations (or satellite derived thickness like Millan et al. 2022) should be made for a benchmark of glaciers to assess errors of using a default A and default sliding parameter (if sliding is also activated in OGGM).

L377-378. Authors could expand their conclusions and use these results to suggest better approaches to simulate albedo in JULES. And expand the very short conclusions.

**Discussion**

The authors should consider validating their initial glacier state (see Major comments).

L483-484. This is an important find and should be included in the abstract.

L486. "wght_alb". Give the full parameter name then refer to the abbreviation in (). Check this throughout the manuscript.

L490-493. Again, another good find that should be highlighted in the Abstract, Introduction and Conclusions.

**Figures and Tables**

Figure 1. It will be nice if in this Figure the authors could add the resolution of JULES grid.

Table 2. Add in the table caption each parameter name description and units.

Figure 4. Add error bars to scatter plots with confidence interval.

Figure 6. Add error bars to scatter plot with confidence interval.

Figure 8. Maybe it is more intuitive for the reader if the authors visualize % of Area change or % of Mass change (e.g., Rounce et al. 2023)

Figure 10. Is too small and very hard to read. I suggest perhaps dividing this in to two figures: Fig 10 for Mean elevation and Fig 11 for Energy Balance.

Appendix C. Specify what are the implications of comparing this study to GloGEM and MAR2012 if this study uses a different glacier inventory, thus a different initial glacier area. Add initial glacier area coverage by each study.

Appendix Figure E. Same problem as Figure 10.

**Code availability**

Specify the OGGM version used in this study and provide a zenodo doi for that version. See OGGM documentation on how to cite the model. https://docs.oggm.org/en/latest/citing-oggm.html

Provide versions and citations for any other python code or tools used.

**References**

Huss, M and Hock, R (2015) A new model for global glacier change and sea-level rise. Frontiers in Earth Science 3. doi: 10.3389/feart.2015.00054

Marzeion, B., Jarosch, A. H., and Hofer, M.: Past and future sea-level change from the surface mass balance of glaciers, The Cryosphere, 6, 1295–1322, https://doi.org/10.5194/tc-6-1295-2012, 2012.

Zekollari, H., Huss, M., Farinotti, D., & Lhermitte, S. (2022). Ice-dynamical glacier evolution modeling—A review. Reviews of Geophysics, 60, e2021RG000754. https://doi.org/10.1029/2021RG000754

R. Millan, J. Mouginot, A. Rabatel, M. Morlighem, Ice velocity and thickness of the world's glaciers. Nat. Geosci. 15, 124–129 (2022).

R. Hugonnet, R. McNabb, E. Berthier, B. Menounos, C. Nuth, L. Girod, D. Farinotti, M. Huss, I. Dussaillant, F. Brun, A. Kääb, Accelerated global glacier mass loss in the early twenty-first century. Nature 592, 726–731 (2021).

David R. Rounce et al. Global glacier change in the 21st century: Every increase in temperature matters.Science 379,78-83(2023).DOI:10.1126/science.abo1324

Li, F., Maussion, F., Wu, G., Chen, W., Yu, Z., Li, Y. and Liu, G.: Influence of glacier inventories on ice thickness estimates and future glacier change projections in the Tian Shan range, Central Asia, J. Glaciol., 69(274), 266–280, doi:10.1017/jog.2022.60, 2023.

Maussion, F., Butenko, A., Champollion, N., Dusch, M., Eis, J., Fourteau, K., Gregor, P., Jarosch, A. H., Landmann, J., Oesterle, F., Recinos, B., Rothenpieler, T., Vlug, A., Wild, C. T., and Marzeion, B.: The Open Global Glacier Model (OGGM) v1.1, Geosci. Model Dev., 12, 909-931, doi:10.5194/gmd-12-909-2019, 2019.

Winkler, M., Juen, I., Mölg, T., Wagnon, P., Gómez, J., and Kaser, G.: Measured and modelled sublimation on the tropical Glaciar Artesonraju, Perú, The Cryosphere, 3, 21-30, 10.5194/tc-3-21-2009, 2009.

Dussaillant, I., Berthier, E., Brun, F., Masiokas, M., Hugonnet, R., Favier, V., Rabatel, A., Pitte, P., and Ruiz, L.: Two decades of glacier mass loss along the Andes, Nature Geoscience, 12, 802-808, 10.1038/s41561-019-0432-5, 2019

---

## Author Response (AR1)

**egusphere-2024-863 response to Anonymous Referee #1 with revisions**

We thank anonymous referee #1 (R1) for their patience. We have now made our proposed revisions to the manuscript based on their comments as well as those of R2. Below, we have kept the original referee comments in black, our original responses in blue and details of our revisions in orange. We have undertaken some restructuring of the text based on R2's comments and some substantial addition to the methods and discussion so, for ease, we have tried to include references to line numbers in the revised text where possible to make it easier to pinpoint our revisions.

The editor also noted that we had not adequately addressed the comment relating to the overestimating albedo and underestimating retreat. This oversight was not intentional and we address this below along with the editor response below.

We thank anonymous referee #1 (R1) for the encouraging and thorough review of our manuscript with clear recommendations. We have considered our response to each comment carefully, especially to those that require clarification or are critical for which we provide more detailed responses with clear justifications. We did find that a small number of comments stem from some confusion about the underlying methodology of this study (e.g. those related the WRF albedo). We take full responsibility for this and plan to be more explicit about some of the details in the revised manuscript. Furthermore, these were still very valuable comments to us and underpin a number of improvements that we propose below. We have included all of the original referee comments in black. Our responses are in blue. Any proposed revisions are written with **bold underlined text**. We look forward to the response from the editor at their earliest convenience.

In this paper, the authors combine the JULES model, which solves the glacier energy balance, with OGGM, which simulates ice flow, and investigate the mass balance over 500 tropical glaciers in the Vilcanota-Urubamba basin in Peru. Specifically, the authors calibrate the surface energy balance model parameters based on 30 glaciers during the period of 2000-2018 and project the mass balance for all glaciers in the region until 2100 with RCP4.5 and RCP8.5. They find that sublimation plays a minor role in glacier evolution at the basal scale, and their mass balance projections are more conservative than previous models. For example, the JULES-OGGM model estimates that 17% of the ice mass will remain by 2100 under RCP4.5, while other models (GloGEM and MAR2012) predict this number to be only 2%.

The manuscript is well written and provides a valuable addition to the current literature of physics-based glacier models, and glacier modeling in general in the region of the tropical Andes. However, I believe that the authors need to provide more information on the climatic input data used in the surface energy balance model. I also suggest that the authors add a sensitivity study for the input data (especially for albedo). Comments and suggestions are given in the list below.

**Major comments:**

**JULES model input parameters:** One of the main challenges when using surface energy balance modeling is obtaining the required input data. The authors mention that the data for the other meteorological variables (other than temperature and precipitation) were "generated by resampling (repeating) the 1980-2018 WRF simulations to produce a continuous 2019-2100 time series" (lines 131-133). By not using predictions from CMIP5 for these variables (radiation, relative humidity, wind speed, etc.), the authors are not using future climate information, but

rather (randomly?) resampling the current climate for those variables. Hence, I am wondering about large biases in their model input data for the surface balance model. Can the authors please elaborate on their reasoning and discuss biases?

We completely agree that establishing suitable climate input data is an important and challenging aspect of any surface energy balance modelling study and we put considerable effort into discussing and developing a justifiable approach to do this. As with the temperature and precipitation variables, it was clear that we could not simply apply the raw CMIP5 climate simulations given the inherent biases in the climate model outputs, in part due to the coarser grid spacing of CMIP5 climate models compared to our WRF model. While statistical downscaling methods, such as the quantile delta mapping approach used in this study, have been applied widely to precipitation and temperature variables, they are not routinely used for other variables such as wind and radiation. This is due to the weaker direct link between these variables across spatial scales. For example, a future increase in meridional near-surface wind in the CMIP5 models does not necessarily translate into a future increase in near-surface wind at the WRF model resolution, due to the influence of topography (amongst others). Similarly, radiation change may depend more on local temperature and humidity than large-scale radiation change. For precipitation and temperature, we had the additional benefit of having a wide distribution of meteorological stations across the catchment to validate our downscaling approaches (more information provided in Potter et al., 2023). The same cannot be said for other meteorological variables which are not routinely monitored in the catchment. On balance, we felt that attempting to downscale the CMIP5 projections of the other variables could introduce additional biases in our input data than resampling our historical, dynamically-downscaled, data. We appreciate, however, that the opposite argument could be made i.e. that by using our historical data, we could be missing important changes in these other variables. Of these, our study indicates that the radiation balance is most important. To explore this aspect in more detail, we extracted the incident long and shortwave radiation data from each CMIP5 model at the point closest to the middle of the domain (at -13.69,-72.02). We found that annual average incident longwave radiation is projected to increase by an average of 6.63 % from the 1980-2018 average to the 2061-2099 average under RCP8.5, with all models showing a statistically significant increase. There is also an increase in average incident shortwave radiation, by 3.20 % over the same time period, with 22 models showing a statistically significant increase, 4 showing a statistically significant decrease and 4 showing no significant change. While these changes are modest, and, as we explain above, do not necessarily translate to equivalent changes at smaller scales, we do feel that, on reflection, we should and **will make the following changes in our revised manuscript: i) better-justify our reasoning for not using all of the climate variable data from the CMIP5 simulations; and ii) provide a more balance and up front consideration of the limitations of this approach (drawing on our analysis of the radiation projections) and the potential impacts on our estimates of twenty-first century glacier retreat.**

We have added additional justification as to why we decided not to downscale the other climate variables in the revised manuscript (**L154**) which largely follows our points above:

*"While statistical downscaling methods like quantile mapping have been applied widely to temperature and precipitation variables in the past, they are not routinely used for other variables such as wind and radiation because of the weaker direct link between these variables across spatial scales. For example, a future increase in meridional near-surface wind in the CMIP5 models would not necessarily translate into a future increase in near-surface wind at the*

*WRF model resolution due to the influence of local scale topography for example. Similarly, radiation change may depend more on local temperature and humidity variability than large scale radiation change. An additional challenge of applying statistical downscaling techniques to the other meteorological variables would have been the lack of validation data as the meteorological stations in the VUB only provide temperature and precipitation variables. For these reasons, we decided not to attempt to statistically downscale the CMIP5 projections of the other variables as this could have introduced additional unknown biases in our input data. Instead, equivalent future data for the other meteorological variables were generated by resampling (repeating) the 1980-2018 WRF simulations to produce a continuous 2019-2100 time series. …"*

We have also included additional discussion points in the discussion section, drawing on our analysis of the radiation data in the discussion (**L620**) with the following text:

*"To explore this limitation in more detail, we extracted the incident long and shortwave radiation data from each CMIP5 ensemble member at the point closest to the middle of the domain (at -13.69ºN, -72.02ºE). They showed that the annual average incident longwave radiation is projected to increase by an average of 6.63% from the 1980-2018 average to the 2061-2099 average under RCP8.5, with all models showing a statistically significant increase. They also showed an increase in average incident shortwave radiation by 3.20 % over the same time period with 22 models showing a statistically significant increase, 4 showing a statistically significant decrease and 4 showing no significant change. While these changes would only result in a relatively small change to the radiation balance, they are not accounted for in our projections."*

Please also provide more information on the resampling procedure.

Yes, we appreciate we could have provided more information on this. To be clear, we repeat the 1980-2018 WRF simulations i.e. we preserve the year-by-year sequencing of the climate variables. We preferred this approach rather than randomly resampling the data to preserve any multi-year cycling e.g. ENSO which may be present in the driving data. We also manually checked the data to ensure that this approach would not introduce any strange "jumps" in the driving data across the 2018 to 1980 crossover and we were satisfied that it would not. **We will provide additional information and justification on our resampling procedure in the revised manuscript.**

We have added the following text, which follows on from our justification of the downscaling approach to provide additional clarity of the resampling method (**L164**):

*"By resampling in this way, we preserve the year-by-year sequencing of the climate variables. This approach was deemed preferable to randomly resampling the data as it preserves any multi-year cycling e.g. ENSO which may be present in the driving data. The data were manually checked to ensure that this approach did not introduce any strange jumps in the driving data across the 2018 to 1980 crossover."*

Have the authors performed a sensitivity study on the input data?

We've answered this question below under "Albedo modeling" where the same question is asked.

**Albedo modeling:** The authors mention a poor performance in the WRF albedo representations (lines 363-375). In particular, WRF largely overestimates the albedo. Hence, it is not surprising

that the JULES-OGGM model shows more conservative mass balance projections for 2100 than other models (e.g., GloGEM, MAR2012, Figure C1).

This appears to be a source of confusion which is also picked up in one of the minor comments below. To be clear, the albedo simulations that we're referring to here are those from JULES, not WRF. We only take the incident radiation variables from WRF. **In the revised manuscript, we will clarify the text in this section to make it clear that we're talking about JULES, not WRF.**

Having re-read the manuscript again, and looked at this section in detail, we do feel that the distinction between WRF and JULES is clearly defined in the manuscript and we have struggled to find ways of making this any clearer. As such, we have not made any alterations focussed on this point specifically.

With regards to the editor's comment where they noted that, "There was indeed a misunderstanding from the reviewer as, according to your response, the albedo is generated by JULES, not by WRF, but this does not diminish the problem of poor albedo simulation you report in the results. Substantial overestimation of albedo, for example, will result in substantial underestimation of glacier melt provided that other variables stay the same. Your current response does not address this criticism adequately as the comment was about the albedo overestimation that you report and not about whether the albedo is generated by WRF or JULES." We agree that we did not address this comment adequately. The point made by R1 is an important one and, as such, we have included this point right at the top of our list potential sources of differences between the JULES-OGGM simulations and those of the MAR2012 and GloGEM in the discussion (**L637**):

*"Arguably of major importance, is the fact that JULES-OGGM was shown to overestimate surface albedo, at least at the one observation point in the basin. If this behaviour is indicative of basin-wide simulations, the apparent conservativeness of the simulated retreat likely stems from this model deficiency."*

Note, that in addition to this, in response to one of the minor comments below and those from R2, we have also expanded the discussion of this limitation specifically in section 4.1 (**L555**).

However, the authors implemented many improvements to the glacier modeling procedures in comparison to previous studies (lines 524-528). Hence, I find it hard to estimate whether the more conservative estimates stem from the problematic albedo modeling and resampled model input data, or whether these are actually more realistic estimates for glaciers in this region. Have the authors performed sensitivity tests?

We appreciate this very thoughtful line of enquiry. Like them, we are also fascinated by the reasons for the more conservative estimates and whether these are more realistic than estimates from other models. We're quite sure, though, that the differences are not simply a result of the problematic albedo modelling and our approach to resampling the other climate variables. The referee may be aware of a recent study by Aguayo et al. (preprint) who compare the sources of glacier projection uncertainty in the Patagonian Andes using OGGM. They show that the historical climate data used to calibrate the model is actually a more important source of projection uncertainty than the future climate. They also show the significance of other aspects such as ice thickness and initial glacier coverage. This study suggests that these additional differences between our model setup and those of GloGEM and MAR2012 are also likely to be important and we allude to this in the discussion (lines 525-531):

*"These differences likely stem from a range of sources in addition to differences in the glacier models themselves. For example, we used statistically-downscaled and bias-corrected climate projections rather than the raw CMIP 5 data. We also used different initial glacier coverage and thickness estimates based on the more refined maps of Drenkhan et al. (2018). Perhaps most-significantly, we tuned JULES-OGGM to geodetic data, while the GlacierMIP models were tuned to World Glacier Monitoring Service (WGMS) data. Indeed, this could explain why, even though the model evaluation showed JULES-OGGM to overestimate mass loss by a factor of two, it still shows to be more conservative than the GlacierMIP models."*

We have consciously remained ambiguous on the exact sources of these differences as we appreciate that the reasons are likely to be multifaceted and a decomposition of these sources (e.g. through sensitivity analysis) is beyond the scope of what is already a comprehensive study. We don't see this as a negative of the paper, but rather something to inspire others to delve deeper into these unknowns. **However, we do feel that we could do much more in the discussion and conclusions to emphasise this as an important focal point for future work. We will do this in our revised manuscript.**

We've now included additional text, drawing on the findings of Aguayo et al. (2023) and others, which at the time of writing has now been accepted for publication, to emphasise the multitude of factors that likely contribute to the differences between the JULES-OGGM simulations and those from the GlacierMIP experiments (**L639**):

*"…we did not use the raw CMIP 5 climate projections used to force the GloGEM and MAR2012 projections. Instead, we used a combination of statistically-downscalled and resampled driving climate data which will have inherently different biases and trends over the twenty-first century simulation period. Aguayo et al. (2023) compared the impact of perturbing various aspects of the model setup for OGGM on glacier runoff simulations for the Patagonian Andes and showed that the uncertainty in simulated glacier runoff could mainly stemmed from the choice of historical climate data used to calibrate the model for 78% ± 21% of the catchment area. In contrast, "future" sources of uncertainty (choice of climate model, emission scenario and bias-correction method) were only the main sources of uncertainty for 18% ± 21% of the catchment area. Similarly, Li et al. (2022) showed that for central Asia, the projected mass loss difference between two identical glacier dynamics models with different initial glacier inventories was higher than that of adjacent emission scenario forcing data.*

*Aguayo et al. (2023) also showed the significance of other aspects model setup such as the initial ice thickness and glacier outline maps. Indeed, our study used the more refined glacier outline maps of Drenkhan et al. (2018) than those in the GlacierMIP and there are clearly differences in our initial glacier areas which we estimate as significantly smaller (186.5 km2) compared to those from the GlacierMIP models which are based on the less-accurate RGI 6.0 glacier outlines (250.1-264.3 km2)."*

We then go on to emphasise the need to decompose the different sources of uncertainty using physically-based models like JULES-OGGM (**L663**):

*"A crucial component in understanding how to improve physically-based glacier evolution models like JULES-OGGM will be developing a fuller understanding of where the sensitivities (and potential sources of error) in the model setup lie. Implementing a framework akin to Aguayo et al. (2023) using JULES-OGGM should be a priority for developing this understanding. One should not expect the relative importance of different sources of uncertainty to be the same."*

And to specifically deduce the sources of discrepancy between models, we've reworded our call to include JULES-OGGM in intercomparison studies:

*"Including more complex models like JULES-OGGM in large-scale model comparison studies like GlacierMIP where input and forcing data are standardised would also help to deduce sources for model discrepancies."*

**Minor comments:**

Abstract: Please mention the grid spacing of your model in the abstract.

An interesting request! This is tricky as the model doesn't strictly have a grid spacing. We hope that this is clear from our explanation of the model setup in section 2.4.3 and Figure 3 where we note that:

*"Each glacier is represented by N ice-covered JULES grid boxes".*

These do not represent a strict "space", but rather the surface energy and mass balance at an elevation. We make this clear by stating that:

*"The outputs from JULES, therefore, indicate how surface mass balance changes as a function of elevation only on each glacier."*

These are used to downscale to the scale of the one dimensional OGGM flowline nodes, which themselves are spaced differently for each glacier:

*"For each OGGM node of that glacier, the mass balance at the OGGM node elevation is extracted from the JULES grid box simulations (yellow dash arrows linking, Figure 3b and d), using linear interpolation to downscale to the resolution of OGGM."*

It is therefore not possible for us to quote a particular grid spacing for JULES-OGGM.

Page 8 line 148-151: Do Dussaillant et al. (2019) provide yearly or seasonal data? Please specify.

They actually provide the data aggregated over a number of years. **We'll make this clear in the revised manuscript.**

We've included an additional sentence here (**L129**) which states that:

*"These data are provided as mean annual elevation change rates over the 18-year time period which were converted to mass loss with an assumed ice density of 917 kg m$^{-3}$."*

Page 8 lines 165-170: Can the authors provide details on how the turbulent and latent heat fluxes and the ground heat flux were calculated?

We appreciate that we could provide more detail on the individual model fluxes, but we purposely decided not to do this, in part for conciseness and in part because the full list of equations are available in Best et al. (2011) which we refer the reader to in lines 155-156. We would prefer not to include these additional details as the manuscript if already quite comprehensive. Instead, we **propose to explicitly refer the reader to the Best et al. (2011) study in the descriptive text of the model energy balance (lines 166-170).**

Having read the manuscript again, we noted that we do in fact already refer the reader to the Best paper for a more detailed explanation of the model in the opening paragraph of this section. Accordingly, we have not added this proposed additional reference to the Best paper.

Page 5 lines 121-122: "Grid spacing" and "resolution" refer to two different length scales and should not be used interchangeably (e.g., Grasso, 2000; Stull, 2015). It would be more appropriate to use "grid spacing" here and for similar cases.

Agree, **we will remove all references to "resolution" where they are not appropriate.**

We noted five uses of resolution, four of which were not appropriate. We have modified all of these in the revised manuscript.

Page 6 section 2.4.1: Can the authors please add some details on the start dates for summer and winter periods in the model?

We're a bit confused by this comment. What do you mean by "in the model?" You can't specify a start date for a season in the model. And how does this relate to the text on page 6 or in section 2.4.1? Could the referee please clarify this for us.

Page 11 lines 225-226: How did the authors come up with 10 grid box nodes with an equal spacing of 167 m elevation for an elevation difference of 2500 m between zmin and zmax? Please explain.

Good spot! It should say 278 m not 167 m. **This number will be corrected in the revised manuscript.**

We've changed this in the revised manuscript (**L273**)

Page 11 lines 236-238: Please provide details on the adjustments that were used for the shortwave radiation.

Yes, on reflection we could have provided more detail here on what we mean by "adjusted". To be clear, we're accounting for the fact that the WRF incident shortwave radiation flux is for a flat horizontal surface whereas the glaciers have a tilted surface. The amount of direct shortwave radiation (i.e. solar radiation received from the sun without having been scattered by the atmosphere) received on a tilted surface will be different to a flat surface. The ratio of the direct incident shortwave radiation received on a tilted surface with respect to a horizontal surface can be calculated from the geometric factor, $R_b = \cos\theta / \cos\theta_z$ , where $\theta$ is the angle of incidence (i.e. the angle of the solar beam relative to the tilted glacier surface), and $\theta_z$ is the solar zenith angle. We calculated hourly $R_b$ over the simulation period using the known glacier slope and aspect (taken from the SRTM digital elevation model) and hourly simulations of the position of the sun in the sky which was calculated using the Pysolar python package. This allowed us to scale the direct incident shortwave radiation dynamically based on the position of the sun in the sky and individual glacier geometries. Note that the WRF incident shortwave radiation flux includes both direct and diffuse (scattered) shortwave radiation. By default, JULES assumes that half of the incident shortwave radiation is direct. As we outline in the manuscript, we use the default parameterisation of JULES except where we have explicitly stated otherwise. There was no clear justification from straying from this for this study. Accordingly, the adjustment was only applied to half of the shortwave radiation flux provided by WRF. **We will include this additional detail in the revised manuscript.**

We've now included this text in the methodology in the revised manuscript (**L194-202**)

Page 11 line 240: typo

Agreed. **We'll address this in the revised manuscript.**

Corrected

Page 11 lines 247-248: Have the authors used 10 JULES grid boxes for every glacier or used less grid boxes for glaciers that don't span the whole range of zmax-zmin?

Yes. **We'll clarify this in the revised manuscript.**

We've added this clarification to the revised manuscript (**L238**):

*"All 10 JULES grid boxes were used for every glacier to ensure that the glacier hypsometry, regardless of its trajectory over the simulation period, was always adequately covered by the JULES simulations."*

Page 12 lines 301-303: Are the results (section 3) based on one set of parameters for all glaciers, or was one set of parameters chosen per subregion (R1-R10)? Please specify.

The latter. We do feel that we have already made this clear in the manuscript and so propose not to change the text here. Lines 318-319 state that:

"*this regional approach was adopted for the remainder of the study*"

Page 16 lines 356-362: Have these two glaciers been part of the 30 glaciers used for calibration?

Yes. **We'll make this clear in the revised manuscript.**

We've now added the additional sentence to the revised manuscript (**L411**):

*"Suyuparina and Quisoquipina glaciers are two of the 30 calibration glaciers."*

Page 17 pages 363-375: WRF albedo modeling: The authors observed that the WRF-modeled albedo rarely falls below 0.8, but the observed albedo falls as low as 0.2 by the end of the dry season. The authors are using the WRF setup from Potter et al. (2023), who are using the Noah-MP land surface model. The default albedo parameterizations for land ice in Noah-MP are set relatively high and might need to be lowered (variable ALBICE in phys/module_sf_noahmp_glacier.F) for a value more consistent with bare ice in the study region. Have the authors explored changes in the WRF land surface model for a more realistic representation of albedo?

We address this in our response above, but to reiterate, we are not referring to the WRF albedo, only the JULES albedo.

Page 21 Figure 9 (e) and (f): Please specify which year you are referring to here (2020?)

Yes, we should have been clearer about this. We're using our initial area data (year 1998) derived by Drenkhan et al. (2018). **We'll make this clear in the revised manuscript.**

We've reworded the caption of the figure to clarify this:

*"spread of glacier area based on initial mapped areas for the year 1998 (e) and initial elevations based on SRTM elevation data for the year 2000 (f)".*

Page 23 Figure 10 and page 33 Figure E1: These figures are hard to read. Please increase the text and line sizes.

We agree, **we'll increase the text size for this figure in the revised manuscript.**

Done. We've had to change the aspect ratio, legends and labelling to make this work, but we feel that the figure is much easier to interpret now.

Page 23 lines 469-470: Can you provide an error estimate of the geodetic validation data used in this study?

Unfortunately, not. Although the geodetic dataset developed by Dussaillant et al. (2018) includes error estimates, these are provided on a glacier-by-glacier basis using the RGI outlines. We used different glacier outlines, which meant that we had to calculate the geodetic mass balance of each glacier ourselves from the 30 m gridded data. To calculate the uncertainty, we'd need the raw analysis data that the authors used at each grid cell. This would be considerable work for something which we feel is desirable, but not fundamental to our study.

Page 24 lines 484-486: Can the authors give a brief overview of the current snow albedo routine in JULES?

Yes, while we did include a very brief overview of the processes included in this routine (lines 171-173), we appreciate that we could expand this slightly to provide additional details relevant to the study. **We'll add this text in the revised manuscript.**

We have now included some additional text here to provide more details on the snow routine (**L555**):

*"The prognostic snow albedo routine in JULES is based on the Wiscombe et al. (1980) spectral snow model which simulates changes to snow albedo, both in the visible and infrared wavelengths as a function of effective snow grain size. The models of snow grain size development and spectral albedo are, in essence, empirical and validated against a small number of observation studies in the Antarctic and Canadian Archipelago…"*

We also use this as a basis for stating some of the known limitations:

*"They also highlight two limitations of the model in the inability to account for: i) snow impurities; and ii) scattering properties of non-spherical snow grain shapes."*

Which we then go on to discuss in more detail.

Page 24 lines 491-496: Getting the net radiation correct in glacier modeling is a (main) challenge beyond tropical glaciers. Are there any conclusions for glacier calibration that can be drawn which are specific to tropical glaciers?

We agree, this is a challenge beyond tropical glaciers, but is one that clearly stands out as fundamental in our study, and therefore warrants it's place in our discussion. We feel that we have been rightly cautious about the transferability of our findings to other regions outside of the tropics and/or generalising our findings across all tropical glaciers or drawing conclusions about tropic-specific characteristics, given the scope of the study (e.g. spatial coverage, validation data, modelling approaches etc). Indeed, in our conclusions we make this clear by stating that:

"*results are not necessarily indicative of all glacierised basins in the tropics*"

and that:

"*we believe there is much to be learned from applying a physically-based, globally-capable model like JULES-OGGM to other basins inside and outside of the tropics*"

Even so, we do draw conclusions that are certainly of interest for modelling tropical glaciers in similar high-altitude, semi-arid settings such as those around the significance (or not) of sublimation processes.

Page 25 lines 526-527: I believe it is important here to mention which variables have been downscaled (i.e., temperature and precipitation), and that the other variables have been resampled for 2019-2100.

We completely agree and, as we mentioned in our response to the major comments, **we will add additional consideration of these aspects of the model setup and their implications for the results that we have presented in the discussion.**

Yes, addressed in our response and revisions detailed above.

References:

Grasso, LD. (2000). The Differentiation between grid spacing and resolution and their application to numerical modeling. Bulletin of the American Meteorological Society. 81 (3). 579-580. 10.1175/1520-0477(2000)081<0579:CAA>2.3.CO;2.

Stull, R. B. (2015). Practical meteorology: An algebra-based survey of atmospheric science. Department of Earth, Ocean & Atmospheric Sciences, University of British Columbia, Vancouver, BC. https://doi.org/10.14288/1.0300441.

References:

Aguayo et al., preprint, https://doi.org/10.5194/egusphere-2023-2325

Best et al., 2011, https://doi.org/10.5194/gmd-4-677-2011

Drenkhan et al., 2018, https://doi.org/10.1016/j.gloplacha.2018.07.005

Dussaillant et al., 2019, https://doi.org/10.1038/s41561-019-0432-5

Potter et al., 2023, https://doi.org/10.1038/s41612-023-00409-z

**egusphere-2024-863 response to Anonymous Referee #2 with revisions**

We thank anonymous referee #2 (R2) for their patience. We have now made our proposed revisions to the manuscript based on their comments as well as those of R1. Below, we have kept the original referee comments in black, our original responses in blue and details of our revisions in orange. We undertook some restructuring of the text based on R2's comments and have also added a substantial amount of text to the discussion section so, for ease, we have tried to include references to the line number in the revised text where possible to make it easier to pinpoint our revisions.

We thank anonymous referee #2 (R2) for the encouraging and thorough review of our manuscript with clear recommendations. We have considered our response to each comment carefully, especially to those that require clarification or are critical for which we provide more detailed responses with clear justifications. We are largely in agreement with their recommendations and have, therefore, proposed revisions that we feel will result in an improved manuscript. We have included all of the original referee comments in black. In some cases, where multiple comments can be addressed with one response we've boxed these comments. Our responses are in blue. Any proposed revisions are written with **bold underlined text**. We look forward to the response from the editor at their earliest convenience.

In this study, the authors present a framework to sequentially couple the ice dynamical part of the Open Global Glacier Model (OGGM) with the full energy balance model for snow and ice from the Joint-UK Land Environment Simulator (JULES). The authors apply this sequential (offline) coupling to 500 glaciers in the tropical Andes and make projections of glacier mass loss until 2100 for different RCP scenarios. They conclude that under RCP4.5 17% of the ice mass will still remain by 2100 which is more than what other glacier modelling studies have predicted e.g., compared to 2% as predicted by GloGEM (Huss and Hock 2015) and Marzeion et al. 2012.

The authors present a very unique and clever workflow which consists of running both models separately; JULES is in charge of computing the annual specific mass balance over discrete points in the study domain. Thus, this model comes up with different relationships of surface mass balance as a function of height per year $SMB(z_i)t$ for each glacier. OGGM then extracts the annual specific mass balance at a particular location, elevation and time. The ice dynamical flowline module of OGGM then inverts for the ice thickness via mass conservation and the Shallow Ice approximation (SIA). For the glacier evolution OGGM solves the continuity equation with the updated SMB distribution given by JULES.

The authors only calibrate parameters within JULES full energy balance model. Ice dynamical parameters in OGGM are not calibrated. JULES SMB calibration consists of several steps where JULES parameters are modified to achieve the best fit to geodetic mass balance data for a benchmark of 30 glaciers; then parameters are extrapolated to other glaciers within small subregions.

Overall, the manuscript is well written and the methods and discussion section has a clear narrative and description of model experiments. It is clear that the authors have put a lot of effort in model calibration and evaluation of SMB. However, some parts of manuscript lack order and could be re-arranged slightly to enhance the impact of the results and discussion section. There are certain aspects of the methods that require clarification and the discussion section neglects to highlight certain limitations of not updating changes in glacier geometry in

JULES simulations. Also, the authors do not asses the implications of not calibrating ice dynamical parameters in their simulations.

I will definitely recommend the publication of the manuscript after the authors clarify some of my questions below and make some changes to the manuscript. I also recommend below how the authors could evaluate their framework limitations regarding ice dynamics.

Major comments:

• The JULES model is unaware of changes in the glacier hypsometry through time i.e., JULES is not aware of glacier retreat simulated by OGGM. Glacier retreat might affect the $SMB(zi)t$ relationships that JULES outputs; if at a specific height $zi$ and location that OGGM node becomes ice-free in a given year. Authors argue that changes in glacier hypsometry through time are accounted by lowering the ice surface and feeding into OGGM an SMB at a lower elevation. How this is decided between timespans and how authors deal with the transition between ice and ice-free areas is not clear. For example, given two points along the flowline, the $SMB(zi)t$ relationship might not hold if the second point (at a lower elevation) becomes an icefree area. Given that ice free physics are qualitatively different than ice covered areas the interpolation might no longer hold. Limitations regarding this issue are not explained by the authors and this should be clarified in the explanation of step 2 in section 2.4.3. and the discussion part.

We completely agree, this is indeed a limitation of the approach that has the potential to introduce discrepancies in surface mass balance for ice free nodes in OGGM and we did not outline this in the text. OGGM updates the glacier geometry on an annual basis. For ice free nodes, it considers ice flowing into the node upstream (when the glacier is growing) and the surface mass balance. So, these discrepancies have the potential to influence the evolution of the glacier extent over time. If we refer to the JULES model used in this study (always ice covered) as JULES$_{ICE}$ and the hypothetical JULES model that is aware of hypsometry changes as JULES$_{HYP}$ , then we can define where discrepancies between the two models and resultant changes in the ice geometry are likely to be most pronounced. For a given annual timestep, this will be on ice free nodes in OGGM where:

1) One of the models simulates a positive annual mass balance while the other simulates a negative or neutral mass balance
2) Both JULES$_{ICE}$ and JULES$_{HYP}$ simulate a positive annual mass balance, but of different magnitude
3) Where the glacier is growing (i.e. ice is flowing in from upstream) and where the simulated annual surface mass balance of JULES$_{ICE}$ and JULES$_{HYP}$ are different

In the fourth scenario, where the glacier is not advancing and where both models simulate a neutral or negative mass balance, JULES$_{ICE}$ and JULES$_{HYP}$ will have no discernible effect on the simulations as both would result in the continuation of an ice-free surface in OGGM.

Given the above and the dominant negative mass balance of the glaciers over the simulation period, we would argue that discrepancies introduced by this limitation are likely to be small in this study. **Regardless, we agree that it is important to highlight this limitation in the text, particularly for readers who may be considering applying the approach to their own case studies. We will do so in the revised manuscript**.

We have now included the following text in the methods sections (**L291**) to highlight this limitation up front to the reader:

*"One important consideration in the presented JULES-OGGM coupling is how the model deals with the varying physics of surface energy fluxes on ice-free and ice-covered regions as a glacier advances and retreats in OGGM. This could be important, particularly when a glacier is advancing, as OGGM considers ice flowing into ice free nodes in combination with surface accumulation and ablation processes when calculating the redistribution of ice mass into previously ice-free regions. It was decided to ignore these differences, in part because including an ice-free representation in JULES would effectively double the computational costs, but also because all evidence suggests that glaciers in the VUB are and will continue to be in a retreat phase over the planned simulation period which would render any differences in ice-free and ice-covered physics inconsequential. The implications of this simplification for this study and for other studies will be explored further in the discussion."*

As suggested by R2, we've also included additional text in the discussion by including a new section which specifically addresses the "ice-free" limitation (**L582**):

*"In the methodology, we noted that the JULES-OGGM sequential coupling does not explicitly account for differences between surface energy balance fluxes on ice-free and ice-covered surfaces which could be important when OGGM determines if a glacier will advance into an ice-free node on a given time step. For ice-free nodes, OGGM takes the sum of the surface mass balance and ice flowing into that node from upstream. If this sum is positive, the glacier "grows" into this node. With the presented JULES-OGGM setup, the surface mass balance for the ice-free node is calculated as though it were ice-covered. This means that there could be some situations where JULES-OGGM simulates the advance of a glacier at a given timestep when an identical model that accounts for the ice-free energy balance does not (and vice versa). As a thought experiment we could hypothesise a JULES-OGGM model that has an equivalent set of ice-free JULES gridbox nodes for each glacier spanning the full elevation range of the glacier. We could then force OGGM with surface mass balance simulations that are tailored to the ice coverage conditions at a given point in time. The overall impact that using such a model would have on the simulation is difficult to postulate given that it will depend a lot on how the exposed ground is parameterised in JULES, but we would hypothesise that sensitivity to this model limitation would be most pronounced for i) advancing glaciers where the surface mass balance will impact the ice thickness at the margin during transition from ice-free to ice-covered; and ii) glaciers where simulated mass balance at the tongue is close to zero and therefore, small changes in the simulated mass balance on ice free nodes could decide whether or not a glacier grows at a particular timestep. For glaciers undergoing sustained retreat where the surface mass balance at the glacier margin is consistently negative, the choice of using JULES-OGGM with or without our hypothetical improvement would have no discernible effect on the simulation. For this reason and the fact that the mass balance of the glaciers in the VUB over the simulation period is negative (particularly at the glacier tongue), we would argue that discrepancies introduced by the ice-free limitation are likely to be small in this study."*

- The simulated glacier geometry that results from the interaction between mass balance and ice flow processes should be compared to ice thickness or volume observations. This is important as the initial glacier state can have a large impact on the simulated glacier evolution (Zekollari, et al. 2022). Authors do not validate the initial ice thickness distribution obtained with their JULES-OGGM framework before doing future simulations. They only evaluate the model by comparing simulated SMB to geodetic and in situ SMB observations.

There might not be in situ glacier thickness measurements for these glaciers, however, the authors could check how their simulated initial ice volumes or ice thickness distributions compare to those from Millan et al (2022) at least for glacier wide volume estimates. Millan et al (2022) is a satellite derived ice thickness product and uses, like OGGM, mass conservation and the SIA, these are not in situ ice thickness observations but if compared to model initial model simulations could provide an idea of the calibration error for the ice dynamic part of the workflow. In other words, errors derived by not calibrating ice dynamical parameters in OGGM or by not updating glacier geometry changes in JULES. By looking at Millan et al (2022) Figure 3b seems to be data available for C Vilcanota glaciers. I encourage the authors to make such a comparison as this could strength the findings of the manuscript.

We really like the idea of having an independent source of validation for the glacier geometry. It's worth noting that, for the initial ice mass at least (year 2000), we show that, cumulated over the VUB, our estimate sits between estimates used for GloGEM and MAR2012 as part of the GlacierMIP experiments (Figure C1 c&d). **We don't make this point in this text and so, will do so in the revised manuscript**.

We've included this in our addition to the discussion in relation to the Millan et al. (2022) data, as detailed below.

Having looked at the Millan et al. (2022) study, this looks like a promising basis to do a glacier-specific validation. Having read the manuscript in detail, we would highlight several important considerations of using these data:

1) Their ice thickness estimates would not permit the validation of the initial ice geometry specifically as suggested by the referee. This is because their calculations are derived from maps of ice velocity are estimated from the years 2017-2018. Our simulations are initialised for the year 2000.

2) They note that the "temporal mismatch" between the ice velocity product, digital elevation model and glacier outlines are important sources of uncertainty which can't easily be quantified. An important component of this is the glacier outlines. They use the same outlines as our initial glacier area (for the year 1998).

3) Errors in ice thickness for thicknesses below 100 m (this accounts for 95% of the glacierised area in the VUB according to this study) are assumed to be >50%.

While there are undoubtedly limitations of using these data, they are the current state-of-the-art and, therefore warrant being used in our analysis. We were keen to conduct this analysis, even before submitting our revisions, and so have done so considering this suggestion. In the figure below, the Millan et al. (2022) ice volume data are plotted against the JULES-OGGM simulation for 2018 for all glaciers. As we note above in limitation 1, we can't use these the validate the initial geometry specifically, but this comparison does implicitly validate this as well as the dynamic behaviour of the model over the 2000-2018 simulation period. Most simulations sit close to the 1:1 relationship (black dashed line). The total JULES-OGGM ice volume for glaciers where Millan data are available (not all glaciers in the VUB) is 7.19 km$^3$ whereas Millan et al. (2022) estimate it as 8.16 km$^3$. The lower estimated volume is perhaps not surprising, especially due to much older glacier outlines used to estimate these. **We will include this analysis and discussion of the limitations of the data and modelling approach in the discussion.**

We have included this in the revised manuscript. We first introduce the data in the methodology section (**L133**):

*"We also obtained glacier volume data from Millan et al. (2022) who use an ice motion mapping approach to estimate ice thickness and volume of the world's glaciers. These data are subject to some uncertainty given the temporal mismatch between their ice velocity maps which were recorded for the years 2017 and 2018, glacier outlines based on maps of Drenkhan et al. (2018) and digital elevation model. Nonetheless, they serve as a useful comparison for glacier dynamics modelling studies."*

We then include the analysis in a new results section (**L397**). Note that we have changed the figure to a log scale so that discrepancies for lower-volume glaciers are easier to see. We've drawn from this new analysis in our discussion (**L653**) as part of our consideration of the reasons for differences between our simulations and those of GloGEM and MAR2012:

*"For the initial ice volume, it is important to note that our simulated ice mass for the year 2000 lies somewhere between that of GloGEM and MAR2012 (Figure C 1c and d) which suggests that this is not a principal driver of the differences in our simulations to these models. Certainly, when we compared our ice volume simulations for 2018 against the estimates of Millan et al. (2022) the level of agreement was encouraging, but there were some discrepancies, especially for the smallest glaciers where the JULES-OGGM simulations generally underestimate glacier volume. Even so, there are significant uncertainties in the Millan et al. (2022) ice volume estimates, given the temporal mismatch in input data. They also note that errors in ice thickness (and thus volume), for thicknesses below 100 m, which accounts for 95% of the glacierised area in the VUB, are assumed to > 50%."*

[Figure]

Minor changes:

**Abstract:**

L26-27: "We conclude that this inhibits the robustness of extrapolating the JULES parameters across multiple glaciers". I will remove this from the abstract because this is true for any glacier model, parameters describing specific aspects of each glacier can't be extrapolated to other glaciers. See calibrations sections from Marzeion et. al. (2012) and Zekollari, et al. (2022).

Agreed, **we'll remove this in the revised manuscript.**

Done. We've reworded this slightly to more clearly make the point that we were trying to originally:

*"We show that the JULES-OGGM model can be parameterised to capture decadal (2000-2018) mass changes of individual glaciers, but that limitations of the JULES prognostic snow model prevent accurate replication of observed surface albedo fluctuations and mass changes across all glaciers simultaneously."*

Authors should highlight the societal importance of their work in the Abstract and how important this region is for water availability.

Nice idea. **We'll add a sentence on this in the abstract of the revised manuscript.**

We've now modified the sentence that introduces the study basin to say:

*"JULES-OGGM is applied to over 500 tropical glaciers in the Vilcanota-Urubamba basin in Peru, home to more than 800,000 people consisting predominantly of rural communities with low socioeconomic development and high vulnerability to climate change."*

**Introduction**

L44. Replace The is ... with "That is especially true in..."

A typo! **"The" should be replaced with "This". We will revise this.**

Corrected in the revised manuscript

L49. "Sublimation" add citation and definition as this is the first time the authors mention the concept.

We feel that for the readership of this journal, the term "sublimation" is well-known and so feel that addition a citation and definition would provide additional, unnecessary wording to a manuscript which is already quite substantial.

L49-50: "sublimation can account for the majority of energy consumed for ablation" change to "can reduce the energy available for melting" (clearer and in line with Winker et al. 2009).

We agree, this is clearer. **We'll change this in the revised manuscript.**

Changed

Add at the end of the introduction the key takeaways from coupling these two models.

We're not entirely sure what this comment is suggesting exactly and so would request further clarification from the referee on this point.

**Methods**

My main suggestion here is to change the outline in the following way:

2.1 Study Area. This section should be moved and integrated into the Introduction, here authors could make the point that a lot of people depend on these water resources (see Millan, et al. 2022 figure 3b) and given the little knowledge surrounding this area, the authors work is highly important.

We appreciate the referee's desire for us to include some context with regards to the significance of the study basin for water resources in the introduction. We agree and will **add a brief description of this in the introduction where we first introduce the Vilcanota-Urubamba basin (line 91).**

We've now added the following text to highlight the wider importance of the work:

*"The basin is home to over 800,000 people with mostly traditional livelihoods, low socioeconomic development and high poverty. Water is used for irrigated agriculture and hydropower plants which both rely on year-round runoff from glaciers. The basin also provides drinking water for the densely populated city of Cusco which has experienced severe drought in recent years."*

We've considered the referee's request to put the study basin section in the introduction, but do not feel that this will improve the manuscript scientifically or in terms of its readability. In fact, with regards to the latter, we feel that it is much more desirable to have a separate study basin section that the reader can easily refer to when considering the results of the study. Therefore, on balance, we would prefer not to change this in the revised manuscript.
* * *
Sections 2.2 – 2.3.

Should be a new section called: 2. Input data and pre-processing. There authors should explain all the data input used in the model and the pre-processing stages needed to ingest the data into JULES-OGGM workflow. E.g., The authors used their own Glacier inventory. The processing of these outlines and why they choose those instead of the Randolph Glacier Inventory could be specified in a subsection e.g., 2.1. Glacier outlines. With this format authors could also expand into the analysis done to the climate input data.

2.3 Glacier Mass Balance data: Specify the advantages of Dussaillant et al (2019) vs Hugonnet, R and others (2021)? Most glacier modelling studies use Hugonnet, R and others (2021) to calibrate parameters in the glacier mass balance. E.g. Rounce et al. 2023.

2.4 JULES – OGGM glacier modelling workflow

Here I think authors could make a simple change to make the outline of the methods more organised:

3.      Methods: glacier modelling workflow

3.1     JULES

3.2     OGGM

3.3     Sequential (offline) coupling of JULES-OGGM

This change will make it clear that the coupling is sequential and that both models are not fully coupled but one feeds input to the other.

We really appreciate the referee's thoughts on the structure of the methods section. We put a lot of effort into structing the manuscript in a way that would gradually introduce the reader to the various aspects of the data, tools and methods used – a difficult task given the number of novelties that we include! We appreciate that this approach has resulted in a methodology section discusses data, processing and models interchangeably within the same sections which may be confusing. We also appreciate that there is some repetition in having "JULES" and "OGGM" subsections in both section 2.4 and 2.5 and the above suggestions highlight this to us further. We like the referee's suggestion of keeping the data and models separate for example. While it is hard to commit to the exact changes suggested by the referee above, **we are more than happy to commit to reviewing our methodology section structure with the aim of streamlining it and making it easier to digest.**

We have made substantial changes to this section taking on board all suggestions from the referee and resulting in, what we think is a much-improved overview of the data and methods employed in the study. Note, we have not followed their guidance exactly, but we feel we have appropriately addressed the core of the referee's concerns. The main restructuring is that we have moved/rearranged all of the text in what was section 2.5 "Model setup and initialisation" into either the modelling workflow section or the input data sections. This includes specific information about the model setup as well as details on pre-processing. By doing so, we have removed the repetition of separate JULES and OGGM subsections in sections 2.4 and 2.5 into one single section as suggested by the referee. Specifically we have:

Included a new section in the methodology titled "Glacier outlines, mass balance and volume data" (**L118**), replacing the old "glacier mass balance" section. This new section now includes details of the Drenkhan et al. (2018) outlines and reasons for choosing it over those in the RGI – note this reasoning was already included in the text, although not up front at the start of the methodology section like it is now in the revised manuscript.

Following the recommendations of the referee, we have consolidated the JULES-OGGM approach into a single "JULES-OGGM glacier modelling workflow section" (**L179**). Within this, we include a climate pre-processing section which includes all of the various pre-processing steps such as bias correction, calculation of lapse rates and correction of incident shortwave radiation fluxes. We did try and incorporate this into the climate data section as suggested by R2, but we found this didn't flow very well because many of the steps required refer to specific aspects of JULES which is introduced at the start of section 2.3.

The JULES and OGGM sections (**L206 and L241 respectively**) now consolidate the model background and the details of the model setup.

As suggested by R2, we have also renamed the final section to "Sequential coupling of JULES-OGGM"

L179. Replace "they" with Shanon et. al (2017).

Good idea, **we'll do this.**

Changed in revised manuscript.

L194. Replace "the flowline model" with "ice dynamical flowline model"

Agree, **we'll change this in the revised manuscript.**

Done.

L194. Replace ": in effect," with "– i.e.," ... L199. Point the reader to the Figure 3a.

Agree, **we'll do this.**

Done.

L202-203. Climate data pre-processing details could be moved to the input data and pre-processing section.

As we state above, **we'll review our methodology section structuring and will include this in our review.**

See our more detailed response above

L252. Add citation for the Randolph Glacier Inventory (RGI). Move that part of the workflow to the Input data and pre-processing section (e.g., add a Glacier outlines section).

As we state above, **we'll review our methodology section structuring and will include this in our review.**

Citation has been added and then see more detailed response above.

Here the authors should mention the implications of not using the RGI, which limits the comparison of JULES-OGGM results with previous model estimates that use the RGI. e.g., Li, F., et al. (2023) found that in High Mountain Asia projected mass loss differences between inventories are higher than between adjacent emission scenarios, illustrating the vital importance of high-quality inventories.

Yes, we agree, the glacier inventory is very important which is why we chose to use one which we consider to be superior to the RGI. In terms of the implications of not using the RGI. The main one is, as you mention, the difficulty in comparing our simulations to other studies. We would like to point out that we do state this in the discussion (lines 526-527), but we appreciate that **we could emphasise this point further and refer to the Li paper cited. We will do this in the revised manuscript.**

As part of the overall package of revisions to the discussion section, we've addressed this comment with specific referral to the Li paper (**L647**):

*"…Li et al. (2022) showed that for central Asia, the projected mass loss difference between two identical glacier dynamics models with different initial glacier inventories was higher than that of adjacent emission scenario forcing data.*

*Aguayo et al. (2023) also showed the significance of other aspects model setup such as the initial ice thickness and glacier outline maps. Indeed, our study used the more refined glacier outline maps of Drenkhan et al. (2018) than those in the GlacierMIP and there are clearly differences in our initial glacier areas which we estimate as significantly smaller (186.5 km2) compared to those from the GlacierMIP models which are based on the less-accurate RGI 6.0 glacier outlines (250.1-264.3 km2)."*

This might not be as relevant for the Andes but having an idea of how different they are in total area coverage per glaciers should justify why authors do not use the RGI when comparing JULES-OGGM results to GloGEM and Marzeion et al. 2012 in Appendix C1.

Authors should add a calculation of how different the Area coverage per glacier is between the RGI and their inventory of choice.

We completely agree, we don't have the exact numbers off the top of our heads, but if you look at Figure C1 a, you can see that GloGEM and MAR2012 have an initial area of approximately 250 km$^2$ whereas our initial area is closer to 180 km$^2$: A significant difference! **We will add this into the revised manuscript.**

We've added this as part of the new text in the revised manuscript that was added to address the previous comment (see above).

L257 When describing OGGM equilibrium assumption. Authors should specify the OGGM version used in their study. (e.g., v.1.4) as that assumption is not required by OGGM in the latest version (see model updates and documentation).

Yes, we specify the version number before this on line 208. **We'll make it clear that this assumption is tied to this version.**

We've now added the version number in the revised manuscript so that it's clearer that this assumption is tied to version 1.4.

L268 "while the default parameters for the ice flow component of OGGM were used." Add the implications of this in the discussion and limitations, point to that section.

Yes, we state this as a limitation in the discussion (lines 479-481), but we appreciate that we could have expanded on this more and point to ways forward in addressing this such as the "dynamic spinup" approach implemented in the latest version of OGGM that allows one to better constrain the creep parameter in OGGM. **We will expand on this in the revised manuscript.**

We have included additional text in the discussion section (**L533**) in response to this comment which acknowledges the impact that not calibrating the OGGM parameterisation:

*"This has obvious implications for the simulated ice dynamical response to climate forcing and for the initial ice thickness inversion step, both of which have the potential to introduce errors into the forward projections."*

It then goes on to expand on method that has been used to include the OGGM creep parameter in the calibration routine:

*"The recent work of Aguayo et al. (2023) has shown how the OGGM creep parameter can be included in the calibration of OGGM. They propose a method whereby the this parameter is selected so that the ice thickness inversion step, used to identify the initial ice volume, matches that of an independent set of glacier volume estimates similar to those of Millan et al. (2022). Within their calibration routine, the model is further validated against RGI glacier outlines for the year 2000 by implementing a "dynamic spinup" of the model from 1980. Only when the model can capture the RGI glacier area, the glacier volume estimates and geodetic mass balance data is the calibration deemed a success."*

We also note that additional gains could be made by implementing such an approach but also exercise caution on computational requirements of doing so:

*"Implementing this type of calibration approach with JULES-OGGM is feasible, although comes with its own computational constraints that would not make this straightforward. Even so, it*

*could be that additional gains in model performance are found by considering the ice dynamical parameters in the model calibration.”*

We have also added a point in the conclusions to highlight the potential gains that could be made by including the ice dynamical parameters in the calibration step (**L683**):

*“Furthermore, we have not examined the role of the ice flow component in the identified model deficiencies and so additional improvements could be made by, for example, including the ice flow parameters with the model calibration step.”*

L271 “except for the temperature lapse rate” Add (See Appendix A).

We’ve already referred the reader to Appendix A by this point in the text so we don’t see any need to include this.

L280. Add citation to the Monte Carlo framework used and add citations for other studies which have used the same strategy.

We understand the referee’s desire to see a citation here, but we think this might be because we’re using the term “framework”. On reflection, this was the wrong word to use. We didn’t implement a framework that’s been previously developed and citable. Rather, we would have been better to say that we used the concept of Monte Carlo i.e. repeated random sampling, to establish the optimal parameter set. While we have tried to be transparent about what we did exactly, e.g. by stating that we took *“random perturbations”* of our parameters and assessed the simulations for goodness of fit and used the *“quasi-random Sobol sampling strategy (Bratley and Fox, 1988) to sample the parameter space efficiently”*. A key piece of information that we missed was that we sampled the parameters from a uniform distribution i.e. we made no assumption about “priors” in the parameter space. These pieces of information sum up our method (or framework as we wrongly refer it as). **We will reword this accordingly and add the extra level of detail about how we sampled the parameter space in the revised manuscript.**

We’ve made two small modifications to the text here. We now refer to using *“the concept of Monte Carlo”* rather than a Mote Carlo framework (**L320**). We also now explicitly state that: *“All parameters from sampled from a uniform distribution.”*

L294. “Used across environmental modelling applications”. Add citations.

Sure, **we will add several citations here.**

There are lots to choose from! We have selected one from water quality modelling, groundwater forecasting and solar photovoltaic system analysis for three broad examples (**L335**).

L298. Add safepython library citation, version used.

Good idea, **we’ll add this.**

Done, the authors of the library ask that users cite the Pianosi 2015 paper which we have now done. We’ve also included the version number.

**Results**

L307-308. Fig 4a shows a perfect correlation, I wonder if multiple parameter combinations could achieve the same thing?

Almost certainly, yes! But to be clear, the point of this is not to identify equifinality (there are some nice recent examples of studies that aim to do this such as Schuster et al., 2023), it is to evaluate how the optimal efficiency of the model changes as you regionalise the parameterisation.

**Model Evaluation**

L347. The authors could enhance the discussion and provide an explanation from where the errors come from. This could be down to several limitations in their approach:

i)  The JULES full energy balance model is not aware of changes in glaciers hypsometry through time. Thus, the SMB(z) relationships do not incorporate well ice dynamical feedback from OGGM. In other words, lowering the ice surface might not be a realistic representation of glacier retreat.

ii)  the ice dynamical parameters in OGGM are not calibrated per glacier. Is it known that ice thickness (and ice volume) decreases as a function of the A factor; thus, perhaps the default parameter in OGGM might not be the right value for these glaciers? See Maussion et al. (2019).

iii)  A comparison between the ice thickness obtained with JULES-OGGM vs Glacier thickness observations (or satellite derived thickness like Millan et al. 2022) should be made for a benchmark of glaciers to assess errors of using a default A and default sliding parameter (if sliding is also activated in OGGM).

Yes, this is also something that was picked up by referee #1. They also highlighted possible errors introduced due to deficiencies in the driving climate data which would be an additional point to consider here. As we stated in our response to referee #1, we have consciously remained ambiguous on the exact sources of deficiencies in the model and reasons for differences compared to GloGEM and MAR2012 models as the reasons are likely to be multifaceted. **But we appreciate that we could have expanded on some of these aspects in the discussion and will do so in the revised manuscript.**

Yes, all of the points raised here have been included in our revised discussion. Each of them are addressed in our previous responses to comments above.

L377-378. Authors could expand their conclusions and use these results to suggest better approaches to simulate albedo in JULES. And expand the very short conclusions.

We'd be happy to point to some recent advances in albedo modelling, especially using land surface models like JULES e.g. see the recent work of Hao et al. (2022). I hope the referee will appreciate, though, that we will probably refrain from saying that these are necessarily better. We'd need to experiment with this to find out first! **We'll add this to the discussion and conclusions.**

In response to this, we have significantly expanded the discussion around which aspects of the albedo modelling could be improved (**L555**). In summary, we point the reader to two known limitations of the model – specifically its inability to account for snow impurities and non-sphericity of snow particles:

*"The prognostic snow albedo routine in JULES is based on the Wiscombe et al. (1980) spectral snow model which simulates changes to snow albedo, both in the visible and infrared wavelengths as a function of effective snow grain size. The models of snow grain size*

*development and spectral albedo are, in essence, empirical and validated against a small number of observation studies in the Antarctic and Canadian Archipelago which raises the question of their validity in other settings. They also highlight two limitations of the model in the inability to account for: i) snow impurities; and ii) scattering properties of non-spherical snow grain shapes."*

We then go on to discuss observations of mineral dust obtained from ice cores and postulate its potential impact on snow albedo:

*"Ice cores drilled from the Quelccaya ice cap on the eastern edge of the VUB show seasonal deposition of mineral dust during the dry season when westerly winds facilitate entrainment and transport from the dry Altiplano (Thompson et al., 2013). (Reis et al., 2022) found that between the years 2002 and 2018, the average dry season dust concentration was 6.8 ± 1.8 ppm but could be as high as 23.1 ppm. A limited number of observation studies suggest that black carbon concentrations of less than 1 ppm can reduce snow albedo by several percent, but that the effect of mineral dust is approximately two orders of magnitude smaller (Willeit et al., 2018)."*

We then explore theoretical modelling studies which explore the combined effects of dust and non-sphericity to highlight the potential importance of these:

*"However, recent numerical modelling studies suggest that the impact of mineral dust on snow albedo could be much more significant and is strongly influenced by snow grain size and non-sphericity. He et al. (2019) explicitly simulated the deposition and redistribution of dust through the snowpack as well as the scattering properties of non-spherical snow grain shapes in the stochastic aerosol-snow albedo model (Liou et al., 2014) and showed that, in the visible wavelength spectrum, snow albedo can be reduced by almost 20% when dust concentration is 100 ppm and snow grain radius exceeds 1000 μm although the magnitude of reduction is dampened for non-spherical snow grains. Hao et al. (2023) demonstrated that including these processes in the E3SM land surface model, allowed them to more closely-match observed albedo dynamics on the Tibetan Plateau. However, they highlight that the combined effects of non-sphericity and snow impurities on albedo are complex, nonlinear, and may be negative or positive depending on the approach used to parameterise them."*

We also postulate the potential importance of biological communities for altering snow albedo before exercising caution on the feasibility of including these additional processes:

*"The composition of other impurities e.g., biological communities which could also serve to reduce snow and ice albedo (Hotaling et al., 2021), have to our knowledge not been observed or quantified in the study region. So, while gains could be made through implementing improvements to the prognostic snow albedo routine in JULES, such improvements would need to be underpinned with by appropriate observation data to appropriately parameterise them. Meeting this challenge will undoubtedly require improvements to our process models, but likely also requires us to better-constrain mountain snowfall timing and frequency which has a significant influence on albedo dynamics (Johnson et al., 2020)."*

**Discussion**

The authors should consider validating their initial glacier state (see Major comments).

This is covered in our response in the major comments section.

L483-484. This is an important find and should be included in the abstract.

Agreed, **this will be included in the revised manuscript.**

We have now included this in the abstract.

L486. "wght_alb". Give the full parameter name then refer to the abbreviation in (). Check this throughout the manuscript.

We have provided a description of all parameters with their abbreviations in Table 1 (page 12). But we appreciate that, given that this is quite early in the manuscript, it may help to remind the in the discussion. **We will do this in the revised manuscript.**

Done!

L490-493. Again, another good find that should be highlighted in the Abstract, Introduction and Conclusions.

We don't agree that findings should go in the introduction for obvious reasons, but we can see the merit of **including this in the abstract and conclusions and will do so in the revised manuscript.**

We have now added a short addition to the abstract to highlight that:

*"Only by forcing the model with observed net radiation variables, were we able to capture observed surface albedo dynamics."*

And we also now highlight the take home point in the conclusions:

*"This study suggests that a key challenge in applying "physical" glacier evolution models at the scale of whole river basins in tropical regions lies in our limited understanding and ability to represent surface albedo dynamics."*

**Figures and Tables**

Figure 1. It will be nice if in this Figure the authors could add the resolution of JULES grid.

We agree, this would be nice, but unfortunately, not practical. As we outline in the methodology (and show diagrammatically in Figure 3), each glacier has 10 JULES grid box nodes. The proportion of the glacier that each one of those nodes represents is variable (both in space and time – so not static) and in actual fact, the majority of the glacier surface is driven with simulations interpolated between these grid nodes. We hope that the referee can understand why this would make visualising the JULES "grid" extremely difficult. For this reason, we have chosen not to try and include the JULES grid on this figure.

Table 2. Add in the table caption each parameter name description and units.

The units are already in the table headings and the descriptions are in Table 1. Rathe than re-write the descriptions in the caption, **we'll refer the reader to Table 1.**

We have now included a reference to Table 1 in the caption.

Figure 4. Add error bars to scatter plots with confidence interval.

Figure 6. Add error bars to scatter plot with confidence interval.

We were also very keen to include error bars on these plots and did put considerable effort into assessing the feasibility of this. We'll explain why we chose not to.

The Dussailant et al. (2019) data provide error estimates on a glacier-by-glacier basis based on the RGI outlines. As we mentioned, we used a different set of glacier outlines (Drenkhan et al. 2018), with a better representation of the real ice area and a different discretisation of the ice-covered areas into individual glaciers (the RGI has a total of X glaciers in the VUB, while we have 532). The glacier-by-glacier Dussailant et al. (2019) error estimates can therefore not be used directly to provide error estimates.

The differences in glacier extents also meant that we could not use the glacier-by-glacier mass balance stats from Dussailant et al. (2019). As we outline in the manuscript, we extracted the mass balance estimates from the 30 m gridded data, also provided by Dussailant et al. (2019). They also provide error estimates for each of these grid points in the form of standard deviations. These are useful for assessing errors at individual grid points, but cannot easily be aggregated to estimate errors over specific regions as they would require the assumption of conformity to a formal statistical distribution (e.g. gaussian) across all grid points and inter-grid-point independence. Both assumptions cannot be satisfied. The only way to calculate the error bounds robustly would be through deriving them directly form the raw analysis data (like the authors do in their paper). This would be considerable work and far beyond the scope of our study.

Figure 8. Maybe it is more intuitive for the reader if the authors visualize % of Area change or % of Mass change (e.g., Rounce et al. 2023)

Perhaps for some readers, but this is quite subjective and not fundamental to findings of our study. It's also easy for a reader to calculate this from the graph if they want to and we include % change in our take home statistics in the text. For these reasons, we prefer to keep this graph as is.

Figure 10. Is too small and very hard to read. I suggest perhaps dividing this in to two figures: Fig 10 for Mean elevation and Fig 11 for Energy Balance.

Yes, this was also picked up by reviewer #1. **We'll address this in the revised manuscript.**

We've now modified the layout of this figure in the revised manuscript to accommodate a larger font size which is now much easier to read.

Appendix C. Specify what are the implications of comparing this study to GloGEM and MAR2012 if this study uses a different glacier inventory, thus a different initial glacier area. Add initial glacier area coverage by each study.

Agree, we've addressed both of these points in our responses above.

Appendix Figure E. Same problem as Figure 10.

As per response regarding Figure 10 above.

**Code availability**

Specify the OGGM version used in this study and provide a zenodo doi for that version. See OGGM documentation on how to cite the model. https://docs.oggm.org/en/latest/citing-oggm.html Provide versions and citations for any other python code or tools used.

We agree, **we'll add this information to the revised manuscript.**

We've now added additional text to the code availability section in the revised manuscript (including citations where possible) to state:

*"The source code for OGGM v1.4 used in this study can be accessed via Maussion et al. (2021). The safepython python library version 0.2.0 used in this study (Pianosi et al., 2015) and Pysolar v0.11 are both available to use for free under the GNU general public license."*

**References**

Huss, M and Hock, R (2015) A new model for global glacier change and sea-level rise. Frontiers in Earth Science 3. doi: 10.3389/feart.2015.00054

Marzeion, B., Jarosch, A. H., and Hofer, M.: Past and future sea-level change from the surface mass balance of glaciers, The Cryosphere, 6, 1295–1322, https://doi.org/10.5194/tc-6-1295-2012, 2012.

Zekollari, H., Huss, M., Farinotti, D., & Lhermitte, S. (2022). Ice-dynamical glacier evolution modeling—A review. Reviews of Geophysics, 60, e2021RG000754. https://doi.org/10.1029/2021RG000754

R. Millan, J. Mouginot, A. Rabatel, M. Morlighem, Ice velocity and thickness of the world's glaciers. Nat. Geosci. 15, 124–129 (2022).

R. Hugonnet, R. McNabb, E. Berthier, B. Menounos, C. Nuth, L. Girod, D. Farinotti, M. Huss, I.

Dussaillant, F. Brun, A. Kääb, Accelerated global glacier mass loss in the early twenty-first century. Nature 592, 726–731 (2021).

David R. Rounce et al. Global glacier change in the 21st century: Every increase in temperature matters.Science 379,78-83(2023).DOI:10.1126/science.abo1324

Li, F., Maussion, F., Wu, G., Chen, W., Yu, Z., Li, Y. and Liu, G.: Influence of glacier inventories on ice thickness estimates and future glacier change projections in the Tian Shan range, Central Asia, J. Glaciol., 69(274), 266–280, doi:10.1017/jog.2022.60, 2023.

Maussion, F., Butenko, A., Champollion, N., Dusch, M., Eis, J., Fourteau, K., Gregor, P., Jarosch, A. H.,

Landmann, J., Oesterle, F., Recinos, B., Rothenpieler, T., Vlug, A., Wild, C. T., and Marzeion, B.: The Open Global Glacier Model (OGGM) v1.1, Geosci. Model Dev., 12, 909-931, doi:10.5194/gmd-12909-2019, 2019.

Winkler, M., Juen, I., Mölg, T., Wagnon, P., Gómez, J., and Kaser, G.: Measured and modelled sublimation on the tropical Glaciar Artesonraju, Perú, The Cryosphere, 3, 21-30, 10.5194/tc-3-212009, 2009.

Dussaillant, I., Berthier, E., Brun, F., Masiokas, M., Hugonnet, R., Favier, V., Rabatel, A., Pitte, P., and

Ruiz, L.: Two decades of glacier mass loss along the Andes, Nature Geoscience, 12, 802-808, 10.1038/s41561-019-0432-5, 2019

References:

Drenkhan et al., 2018, https://doi.org/10.1016/j.gloplacha.2018.07.005

Dussaillant et al., 2019, https://doi.org/10.1038/s41561-019-0432-5

Hao et al., 2022, https://doi.org/10.5194/gmd-16-75-2023

Schuster et al., 2023, https://doi:10.1017/aog.2023.57